# Inverse design of metal-organic frameworks using deep dreaming approaches

Conor Cleeton ✉ & Lev Sarkisov ✉

Exploring the expansive and largely untapped chemical space of metal-organic frameworks (MOFs) holds promise for revolutionising the field of materials science. MOFs, hailed for their modular architecture, offer unmatched flexibility in customising functionalities to meet specific application needs. However, navigating this chemical space to identify optimal MOF structures poses a significant challenge. Traditional high-throughput computational screening (HTCS), while useful, is often limited by a distribution bias towards materials not aligned with the desired functionalities. To overcome these limitations, this study adopts a "deep dreaming" methodology to optimise MOFs in silico, aiming to generate structures with systematically shifted properties that are closer to target functionalities from the outset. Our approach integrates property prediction and structure optimisation within a single interpretable framework, leveraging a specialised chemical language model augmented with attention mechanisms. Focusing on a curated set of MOF properties critical to applications like carbon capture and energy storage, we demonstrate how deep dreaming can be utilised as a tool for targeted material design.

Metal-organic frameworks (MOFs) have revolutionised the field of materials science, offering a wide range of applications from photocatalysis[1] and semiconductors[2] to energy storage[3] and gas separations[4,5]. The unique appeal of these 'designer' materials lies in their extraordinary tunability, a feature attributed to their modular construction of inorganic nodes and organic linkers in a topological net. As such, the number of experimentally synthesised MOFs currently spans over 100,000[6]. However, the expansive chemical space of organic linkers hints at an almost limitless possibility for MOF combinations, and therefore functionalities. This capacity for customisation is reflected by the vast number of hypothetical MOF structures that have been proposed to date[7].

High-throughput computational screening (HTCS), aided at times by the use of machine learning (ML), is often used to seek promising MOF candidates for a particular application[4]. This approach, which begins with a collection of structures and filters them for a desired key performance indicator (KPI), relies on the selection of top performers from existing databases of materials[2,7,8]. One of the main limitations of conventional HTCS is that the distribution of known materials is usually skewed towards functionalities far away from the intended application. A notable example is in direct air capture (DAC) technologies, where capturing atmospheric concentrations of $CO_2$ poses a significant engineering challenge. Chemisorbents, which selectively bind adsorbates through chemical bond formation[9], are typically preferred for this process due to their strong adsorption characteristics[10]. However, MOFs as physisorbents for DAC have become an attractive alternative due to their more modest sorbent regeneration requirements[11,12]. In most cases, the interactions with $CO_2$ in these materials are relatively weak (typically less than 30 kJ mol⁻¹), resulting in low $CO_2$ uptakes at the relevant process conditions. It is therefore crucial here (and in many other applications) to offer a significantly expanded selection of potential structures for HTCS efforts. This allows researchers to refine their search for the best materials to synthesise, considering a range of aspects (such as linker availability, ease of synthesis, material costs, and possible stability challenges) that go beyond what traditional HTCS studies typically address.

This realisation has led to a growing body of research interested in engineering new MOFs with desired functionalities in recent years[5,13–16].

Department of Chemical Engineering, University of Manchester, Manchester, UK. ✉e-mail: conor.cleeton@manchester.ac.uk; lev.sarkisov@manchester.ac.uk

That is, given a target KPI, generate a MOF or collection of MOFs in silico which satisfies this KPI. This concept, known as inverse design[17], can be facilitated by several avenues. One such avenue is to optimise the combination of topology and secondary building units (SBUs)—the node and linker building blocks—from a library of predetermined SBUs[16]. The main disadvantage of this approach is that the chemical diversity of the resulting MOFs is limited by the SBUs sampled, which usually represent a very small fraction of the potential chemical landscape[18]. Other efforts in this domain rely on the use of generative models, which automatically generate new materials (with novel building blocks) by leveraging large datasets of material properties and structures to learn underlying patterns and relationships. Different architectural flavours of generative models have been used for MOF design in the past, such as VAEs[13], GANs[19], diffusion models[15], and reinforcement learning[5]. A recurring theme in these studies is the strategy of pairing a structure-generating module with a property-predicting module, where communication between the two is limited by a reward or loss function. For example, in VAEs, GANs, and diffusion models, a structure generator learns to sample molecules or materials that resemble an unlabelled dataset through extensive self-supervised pretraining using hundreds of thousands of data points. The generator is then directed by the property predictor—trained on a smaller subset of labelled data points—through an optimisation process that explores a latent chemical space[13,15]. A similar principle applies to reinforcement learning: a pretrained structure generator (acting as the agent) learns to create new structures based on a reward system provided by a property predictor (serving as the environment), with the objective of maximising the reward[5,20–22]. While effective, these optimisation strategies have so far failed to integrate both property prediction and structure optimising tasks into a single module.

There have been a few attempts to address this gap, mostly within the realm of small drug-like molecules[23–25]. One example is the work of Shen et al.[26], who introduce the concept of 'deep dreaming' for the optimisation of organic molecules. This approach begins by training an ML model to predict molecular properties from their string-based representations using only labelled data. Then, through inceptionism—a technique originally developed to visualise the patterns and features learned by convolutional neural networks from images—the ML architecture is inverted to modify the input towards a target property value, effectively creating new molecules in the process. Shen et al.'s findings reveal a few keys points: first, that a model's learned representations can be used to optimise molecular structures in the real chemical space towards desired functionalities using gradient-based algorithms (rather than optimising molecules indirectly in a latent space or through a reinforcement policy), thereby generating distributions of molecules with systematically shifted properties; second, that inverse design can be achieved in a relatively data-efficient manner by relying solely on labelled datasets to train a regression model, thus eliminating the need for a large-scale pretraining phase; and third, that inceptionism allows one to probe what the model has learned of the structure-property relationship by observing the sequence of molecular modifications executed to achieve a particular design objective. Deep dreaming, therefore, offers a way to expand the pool of possible candidate materials for a given application, consolidating the tasks of property prediction and structure optimisation, while also providing interpretable insights into the model's understanding.

In this work, we extend the approach of Shen et al. to MOFs for linker optimisation, which is the area of MOF design with the highest capacity for customisation and therefore functional tunability. We first develop a bespoke chemical language model, augmented with attention mechanisms, to map string-based representations of MOFs to their property values. Then, we curate a synthetic dataset of MOFs using pormake[16], and train our model using data generated either from molecular simulation or from ML surrogates. As a case study, we explore a variety of MOF properties such as the specific heat capacity ($c_p$ [J g$^{-1}$ K$^{-1}$]), gravimetric surface area (GSA [m$^2$ g$^{-1}$]), void fraction (VF [-]), bandgap [eV], heat of $CO_2$ adsorption at infinite dilution ($Q_{CO_2}$ [kJ mol$^{-1}$]), and the $CO_2/N_2$ Henry selectivity ($S_{CO_2/N_2}$ [-]), which find practical applications in carbon capture processes[4,27–29], energy storage[3,30], semiconductors[2,31], and adsorption-based separations[4,5,32]. Once the model is trained to predict a given property value, the same architecture is used to optimise the linker of a MOF via deep molecular dreaming.

The main objective of our work is to explore deep dreaming as a versatile tool to generate MOF populations with systematically shifted properties, all within the native architecture of the property prediction model itself. By placing special emphasis on the physical interpretability of our model and its ability to rapidly generate MOF structures, we deviate from the pursuit of a singular 'best' material. Instead, this approach aims to position us within a design space that is inherently closer to the desired functionalities from the outset, which establishes a platform for further exploratory studies in MOF research.

## Results

### The deep dreaming experiment

Deep dreaming is an experiment that aims to understand how neural networks learn from data[33]. For example, one can train a model to classify images by showing it millions of labelled examples and optimising the network's parameters to achieve accurate classifications. For a task such as this, the network encodes increasingly complex features across layers, from simple textures to shapes of objects, which allows the model to differentiate one image class from another. Once trained, one can visualise what the neural network has learned in these internal layers by fixing the weights and biases of the model, feeding it an image, and then asking it to enhance certain features. This occurs by the process of inceptionism, whereby the network's parameters are no longer updated to minimise the classification error via gradient descent, but rather the input image is modified by maximising the activations of a particular layer via gradient ascent. As each layer is responsible for different levels of abstraction, this process can produce 'dream-like' interpretations of the original image, aptly leading to the term 'deep dreaming'.

Shen et al.[26] leveraged this idea for the inverse design of small organic molecules. However, instead of learning to classify images, they trained a neural network model to predict real-valued properties of molecules from their SELFIES[34] string representations. To learn from these non-numerical features effectively, the SELFIES strings were one-hot encoded—a process that converts each SELFIES token into a binary vector. This vector has a '1' in the position corresponding to the token and '0' elsewhere, ensuring that each character is uniquely represented. Then, by adding noise to the zero elements, they transformed the input from a collection of binary vectors into differentiable probability distributions over the SELFIES tokens. The rationale for this final step becomes clear in the discussion below.

During training—the forward process—the inputs (noisy one-hot vectors) and outputs (property values) remain fixed while the weights and biases of the ML model are incrementally updated via back-propagation to minimise the prediction errors. In molecular deep dreaming, or the reverse process, this paradigm is inverted: the pre-trained weights and biases of the model are frozen, and the input is incrementally modified towards a new, optimal feature vector using gradient descent. This is achieved by first computing the target error, which is the error between the predicted property of a molecule and some target property. Then, the gradient of the target error with respect to the encoded input is calculated, and the initial molecular structure is modified via backpropagation such that the target error is minimised. Gradient-based molecular optimisation such as this is only possible because the molecule is represented by a differentiable probability distribution over SELFIES tokens. This highlights the importance of noisy one-hot encoding in facilitating deep dreaming

experiments. As SELFIES is 100% chemically robust, the optimised representation will always decode to a valid molecule and, provided the mapping between molecular strings and their properties is sufficiently accurate, will return a structure with improved functionality.

## Deep dreaming for MOFs

We extend the methodology described above for the inverse design of MOFs. Our aim is to recover an optimised representation that corresponds to a structure with enhanced properties. However, this task is more challenging as we need to consider not only the edge SBU (the organic linker with two connection points that bridges between node SBUs), but also the node SBU (the inorganic component with more than two connection points), their interactions, and how they coordinate in a given topology. Each of these elements may contribute to our ML model predictions to varying degrees depending on the property of interest[27,35]. It is therefore important to capture these dependencies simultaneously within our approach. Additionally, it is worth reiterating that our primary goal is to optimise the edge SBU only, and in this work, we will focus exclusively on MOFs that feature a single unique edge SBU. To achieve this, the edge SBU encoding must be differentiable, enabling our model to modify it using target error gradient information.

With these considerations in mind, we utilise a textual representation of MOFs inspired by the MOF identification scheme (MOFid) of Bucior et al.[36] In particular, our MOF string comprises three parts: (1) an edge SBU representation, which is described using Group SELFIES[37] strings to preserve important chemical substructures, such as functional groups and aromatic rings; (2) a node SBU representation, described using SELFIES[34] strings; and (3) a topology representation, described using RCSR codes[38]. Collectively, these elements capture the chemical and some topological aspects of MOFs, as detailed in the 'Methods' section. Note that we do not include any structural details here, like the 3D atomic coordinates or geometrical properties of the MOF. While this approach may be considered less expressive than structure-based models such as graph neural networks[39], it allows for an efficient exploration of the property space through straightforward string manipulations in the reverse process. This strategy, therefore, avoids the significant computational demands associated with generating and analysing hundreds of MOF structures during deep dreaming.

Next, we develop an ML architecture in PyTorch (v.2.2.1) designed for complex sequence-to-regression tasks, as shown in Fig. 1a. The model integrates Long-Short Term Memory (LSTM) networks, augmented with attention mechanisms, alongside conventional multilayer perceptron (MLP) neural networks. To prepare the MOF strings as input, they are first decomposed into differentiable (edge) and non-differentiable (node and topology) components. The edge is encoded using a noisy one-hot vector, similar to the work of Shen et al.[26] (Fig. 1b). Since the node and topology representations remain fixed during deep dreaming, we utilise more advanced token embeddings to encode their contributions[40]. We then process the differentiable and non-differentiable portions of the MOF strings using two distinct LSTM branches. An integral component of this architecture is the softmax self-attention mechanism applied to the outputs of both LSTM branches. By incorporating self-attention, the model can focus on the most relevant parts of the input sequence for the regression task at hand. More importantly, this augmentation enhances the model's physical interpretability because the attention weights, when applied during inference, provide insights into the importance of different tokens within our MOF strings and their contributions to the model's predictions. Finally, the context vectors obtained from both LSTM branches are then concatenated and passed through an MLP to obtain the final property prediction. Additional details on the justification of our model architecture, model hyperparameters, and training can be found in the 'Methods' section.

To reverse-engineer MOFs, we train separate models to learn a mapping, $f(X) \rightarrow \hat{y}$, between MOF strings ($X$) and the properties of interest ($\hat{y}$). For this purpose, we use $10{,}000$, $1000$, and $1000$ pormake-generated MOF structures for training, validation, and testing purposes, respectively (see *Dataset curation* in the 'Methods' section). We achieve a coefficient of determination, $R^2$, for the regression performances ranging from 0.99 (for $c_p$) to 0.69 (for $S_{CO_2/N_2}$) (see Supplementary Note 3). The same architectures are then used to optimise MOF linkers, $f(\hat{y}, y_T) \rightarrow X$, by modifying the edge SBU encodings of a seed structure towards a target property, $y_T$ (Fig. 2). While every transmutation is chemically valid by virtue of the Group SELFIES algorithm, not every decoded molecule represents a valid linker. We consider a linker to be valid if it has two connection points (which we represent using francium pseudoatoms), and these connection points must be sufficiently far away from each other to encourage the formation of a porous crystalline structure. We therefore penalise the formation of molecules with chemical compositions that do not contain two francium atoms, and only retain structures with connection points that are appropriately distanced (see Supplementary Note 4).

## Optimising individual MOF structures

Here we demonstrate the deep dreaming experiment for a single MOF using a material property with intuitive optimisation trajectories: the VF. The VF (among other structural features) are intrinsically linked to the functionality of a MOF[41]. For instance, enhancing the pore space–and consequently the VF–can significantly increase the $CO_2$ working capacity in carbon capture applications or the storage capacity in hydrogen/methane storage applications. This is often achieved by extending the length of linker molecules[42]. By this logic, we can expect that an ML model trained to increase the VF of a MOF would adopt a similar optimisation strategy.

To further explore this idea, let us task the deep dreaming model with maximising the VF of a MOF. We can do this by assigning a target VF which, in this experiment, we will set unphysically high at 1. Starting from a seed MOF structure, the model will make minor adjustments to the edge SBU encoding over hundreds of training epochs corresponding to the deep dreaming process. Every adjustment has the potential to produce a pronounced transmutation of the linker molecule. However, when the model traverses regions of the chemical space that decode to an invalid linker molecule, an additional loss term is added to the target loss in order to guide the optimisation down valid chemical pathways. For example, in Fig. 3, we define the valid transmutation pathway as the series of successful linker modifications, while the transmutation pathway simply refers to all the modifications suggested by the model, valid or not. By inspection of the valid transmutation pathway, we can clearly see an extension between the connection points of the linker from the initial seed to the final decoded structure, as anticipated. Amongst the pool of valid candidates, the model also proposes invalid linker molecules, such as those with the incorrect number of connection points or whose connection points are too close together. These structures are penalised and discarded from consideration.

The example shown in Fig. 3 is the result of a single execution of the deep dreaming model. However, since the edge SBU is initialised with random noise before reverse differentiation, the optimisation trajectory (and therefore the final decoded structure) is entirely stochastic. For instance, in Fig. 4, the same seed edge SBU returns different decoded structures across several optimisation runs. In each iteration, the MOF seed is transformed into a structure with an expanded pore space. This is achieved by adding aromatic Group SELFIES tokens into the edge SBU encoding in some cases (i.e., Run 1–3), or by unfurling the 3-membered ring of the seed linker and extending the alkane backbone between connection points (i.e., Run 40). It is therefore evident that, while the trajectory may change across multiple repeat trials, the design strategy remains consistent.

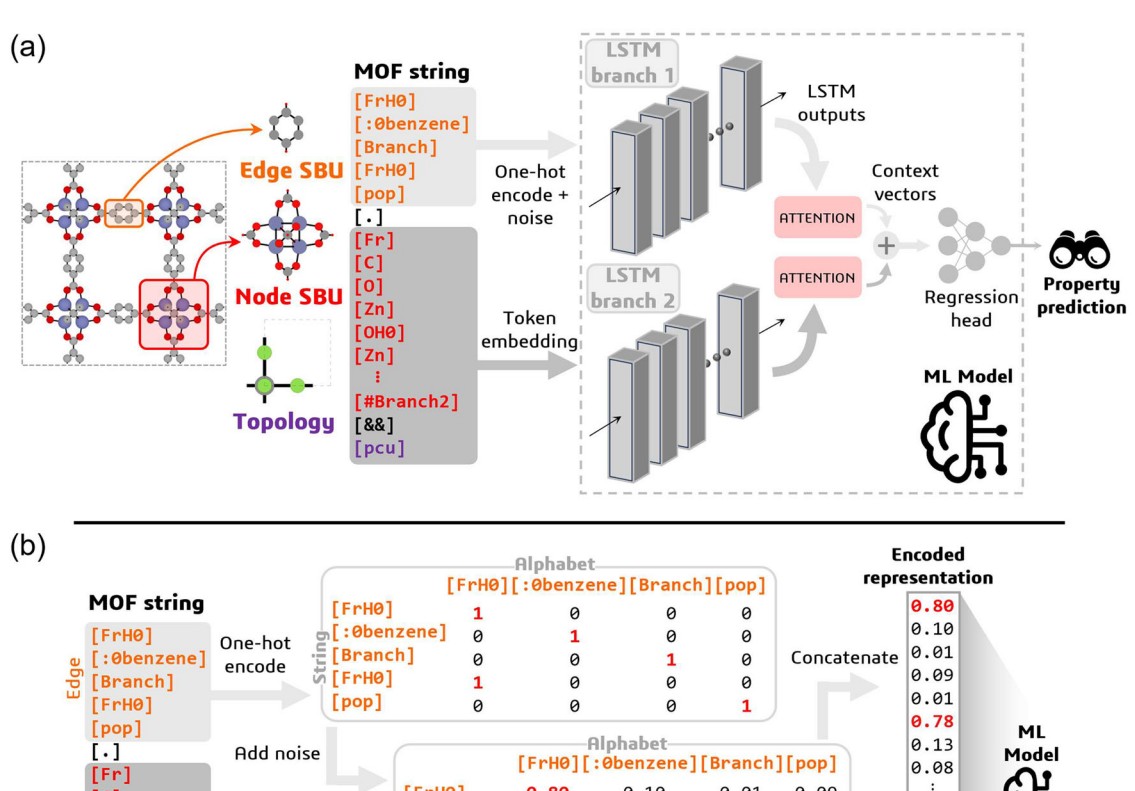

**Fig. 1 | Deep dreaming model and metal-organic framework (MOF) string representation. a** Deep dreaming modelling approach. MOFs are decomposed into an edge secondary building unit (SBU), a node SBU, and a topology, which we characterise using a string-based representation. A bespoke sequence-to-regression machine learning (ML) model, composed of long-short-term memory (LSTM) cells, attention mechanisms, and multilayer perceptrons, is then used to map MOF strings to their respective material properties. To facilitate the optimisation of MOF linkers via reverse differentiation of the input, the linkers are encoded using a one-hot encoding with random noise added to the zero elements.

To capture contributions from the node and topology, we use token embeddings. Connection points between SBUs are identified using francium pseudoatoms in the MOF string representation. **b** Encoding MOF strings. Edge SBUs are represented as SELFIES strings, which can be converted into a machine-readable feature vector by using a one-hot encoding, which represents tokens as binary vectors. Random noise is then added to the zero elements of the one-hot encoded representations, effectively transforming them from a simple bit vector into a differentiable probability distribution over SELFIES tokens. The complete MOF string encoding is then obtained by concatenating the edge, node, and topology encodings.

We can interpret these results as a collection of local optimisations, whereby the ease of transforming the initial structure to achieve the desired functionality may vary based on the chosen optimisation path. Given that the decoded structure obtained by a single run can be highly variable due to the formation of local minima in the optimisation landscape, a more reliable outcome can be achieved by selecting the best-predicted structure from an ensemble of local optimisations. Moving forward, we embrace this methodology, choosing the best structure from an aggregate of local optimisations for any given MOF.

**Interpretable AI**

At this point, it is instructive to highlight one of the important advantages of deep dreaming. While there is no way to deterministically interpolate between structures using this approach, one could argue that the stochastic nature of the deep dreaming process actually imbues it with a high level of physical interpretability. If we examine the individual transmutation pathways as in Fig. 3 or consider the ensemble of decoded structures from the same seed as in Fig. 4, we can garner insight into the model's understanding. For the particular case

of VF, recovering appropriate design principles from the sequence of linker modifications proves to be straightforward and somewhat predictable: a longer linker typically results in a higher VF. This simple correlation manifests both in the individual transmutation pathways and in the collection of decoded structures. However, deciphering such correlations and extracting generalisable insights from ML models can often be complex. One common approach is to provide explanations of the model predictions using feature importance analysis[17]. Translating the outcomes of these analyses into useful design rules depends upon the ability to relate ML features to tangible, physical properties; no amount of post-hoc explanation allows one to engineer new materials if the features themselves cannot be exploited. In our case, by employing MOF strings that directly correspond to chemical structures, we ensure our features are not only physically meaningful but also potentially actionable: an experienced chemist, equipped with the appropriate experimental tools, could replicate the linker modifications suggested by the model. Furthermore, we can (in principle) derive valuable design rules by mimicking the model's automated optimisation strategy. In this capacity, deep dreaming can achieve concurrent objectives in both scientific discovery as well as

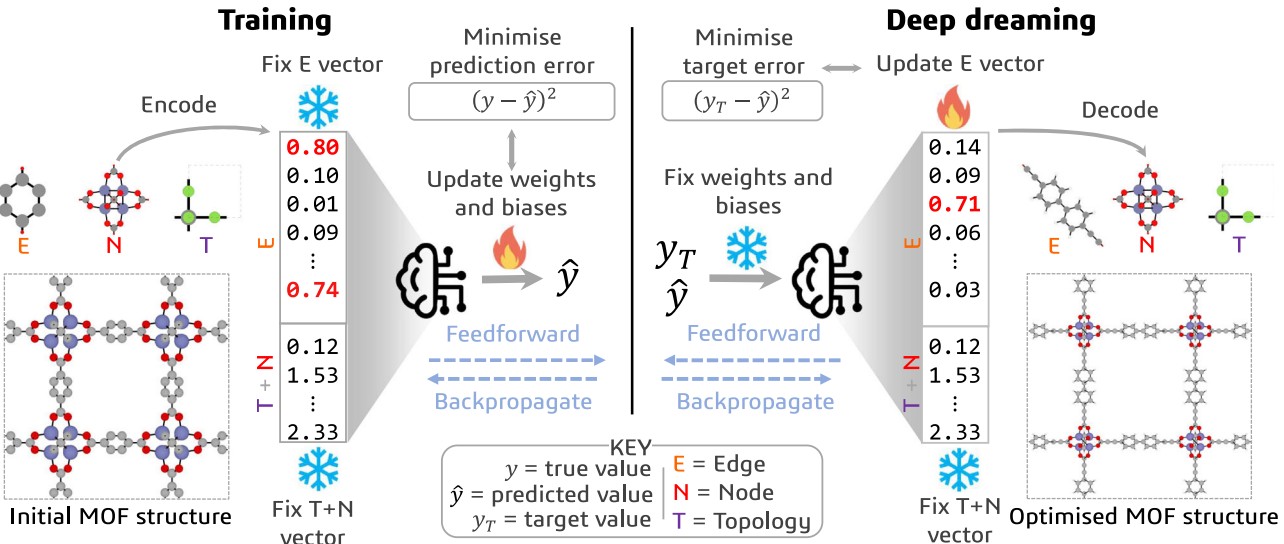

**Fig. 2 | Reverse engineering metal-organic frameworks (MOFs) using deep dreaming.** (Left) In the forward (training) phase, the machine learning model learns to predict the properties of MOFs, $\hat{y}$, from their (fixed) encoded representations by adjusting the model weights and biases through a standard feedforward and backpropagation process to minimise the prediction errors. (Right) In the reverse (dreaming) phase, the same feedforward and backpropagation process is used to update the edge encoding—keeping the weights and biases of the model fixed—to minimise the difference between the predicted property of an initial MOF structure ($\hat{y}$) and some target property ($y_T$). The sequence of modifications executed by the model, from the initial MOF structure to the final optimised MOF structure, forms the optimisation pathway.

understanding[43], thereby shedding a light into the AI 'black box' and moving towards more explainable AI.

## Shifting property distributions

Equipped with the knowledge of how single MOF structures are optimised and their results interpreted, in this section, we examine the model's ability to shift property distributions toward desired functionalities (i.e., optimising entire MOF populations). As a case study, we explore the $c_p$ of MOFs, specifically in the context of adsorption-based carbon capture technologies. For DAC, MOFs are typically considered for deployment in temperature swing adsorption cycles, whereby $CO_2$ is selectively adsorbed at low temperature and then recovered at a higher temperature[9]. The viability of this process crucially depends on minimising the heat required to regenerate the material per kilogram of recovered $CO_2$[27], which is a design objective favoured by MOFs with lower $c_p$. Conversely, pressure swing adsorption cycles, which capture $CO_2$ at high pressure and release it at lower pressures, are preferred for post-combustion carbon capture[4]. The performance of these processes benefits from isothermal conditions, and therefore by MOFs with higher $c_p$[28]. Thus, examining the $c_p$ of MOFs in relation to carbon capture processes provides an excellent benchmark to probe the capabilities of the model in a setting where one might be interested in either minimising or maximising the property value, depending on the target application (however, for completeness, the dreaming results for other MOF properties are provided in Supplementary Note 5).

To that end, from an initial seed distribution of 1000 MOFs, we optimise the $c_p$ by setting the target value high for property maximisation and low for property minimisation. Here, high and low targets refer to the upper and lower limit values encountered in the training set, respectively. Taking only the best-predicted structure from an aggregate of 10 local optimisations, we reconstruct MOFs from their encoded representations without relaxing the crystal structures (see the 'Dataset curation' section) and recompute the 'ground truth' $c_p$ values using the ML model of Moosavi et al.[27]. The results are displayed in Fig. 5a. For both $c_p^{min}$ and $c_p^{max}$, we observe shifts in the property distributions towards the desired functionalities much further away from the central tendency of the seed distribution. These results are shown to be insensitive to the exact atomic

coordinates of the MOFs, as detailed in Supplementary Note 5, where crystal structure relaxation has minimal impact on the property distributions. This successful outcome is facilitated by the design of linkers that closely reflect the physics of $c_p$ itself: in crystalline materials such as MOFs, atoms are arranged in a lattice structure and can only vibrate about their fixed positions. The $c_p$, which measures the energy required to increase the temperature of the solid by one degree, directly correlates to the energy required to alter these vibrations[27]. Notably, these alterations require more energy for heavier atoms, resulting in a higher $c_p$ per atom (expressed in J mol$^{-1}$ K$^{-1}$). However, experimental measurements of $c_p$ are often reported per gram of material, so expressing $c_p$ in units of J g$^{-1}$ K$^{-1}$ offers more practical insights. In this case, linker molecules with a greater molecular weight tend to produce MOFs with a higher atomic $c_p$ but a lower gravimetric $c_p$. Figure 5a reveals that, when tasked with minimising the gravimetric $c_p$, our deep dreaming model favours the addition of heavy atoms like bromine, sulphur, or phosphorus, leading to MOFs with a higher average atomic mass and, consequently, to lower $c_p$ values. This is evident when we inspect the attention weights of the model, whereby Group SELFIES tokens such as `[Br]` and, in the example shown in Fig. 5b, `[pyridine]`, are given more weight relative to their neighbour tokens. On the other hand, MOF linkers optimised for higher $c_p$ are mainly composed of carbon, nitrogen, and oxygen; in other words, by linkers with relatively lower molecular weight. This pattern aligns with observations from the seed distribution, where MOF structures falling within the medium $c_p$ range feature linkers of moderate molecular weight. Just like with the VF, these optimisation strategies are also evident in the individual transmutation pathways, examples of which, along with their automated interpretations, are detailed in Supplementary Note 6.

While our results illustrate the model's nuanced approach to optimising MOF properties through strategic linker modifications, a natural question emerges on the diversity, novelty, and uniqueness of the linkers generated in the reverse process. We therefore evaluate the linkers proposed by our model using the MOSES[44] framework, as described in the 'Methods' section. It is recommended within this framework to benchmark the performance of generative models using 10,000 samples, and so we extend our analysis from Fig. 5a to a

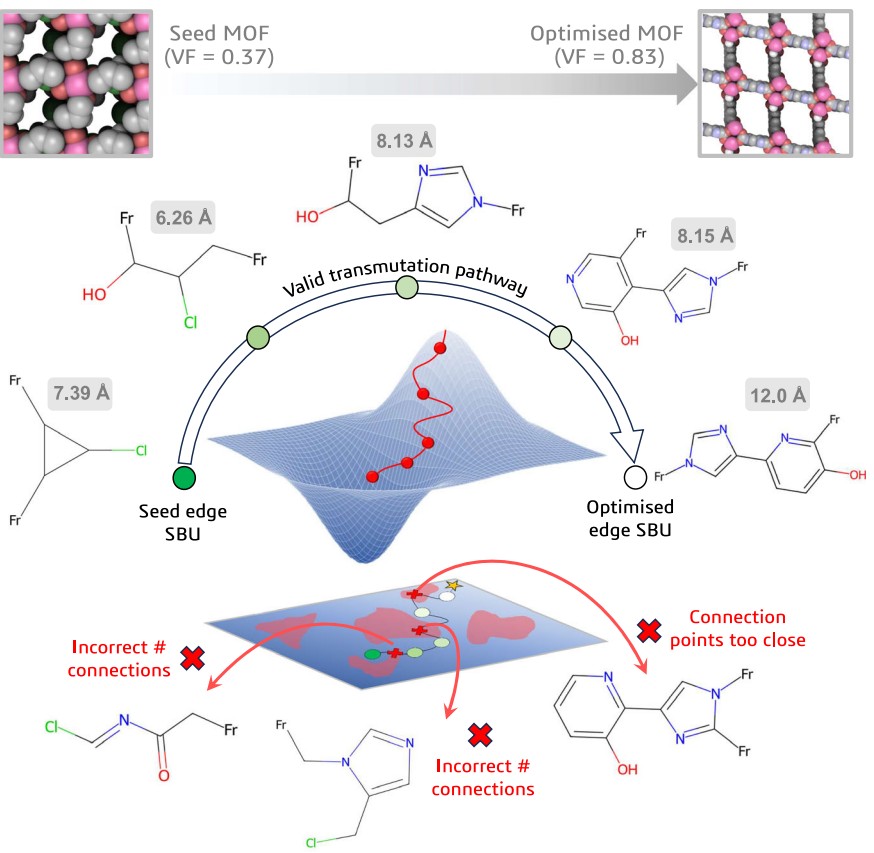

**Fig. 3 | Optimising single metal-organic framework (MOF) structures.** Here, we task the model with maximising the void fraction (VF) of a MOF structure. The seed edge secondary building unit (SBU) is encoded and then optimised using the deep dreaming model. Above each valid linker transmutation, the atomic distance between connection points is shown. As the optimisation proceeds, the length of the linker molecule extends, leading to an increase in the VF of the MOF from 0.37 to 0.83. Along the optimisation trajectory, unsuccessful linker modifications (indicated by the red-coloured patches in the landscape projection) are discarded, such as those with the incorrect number of connection points or whose connection points are too close together. The 3D optimisation contour and its projection are purely illustrative and do not represent an actual material property landscape.

distribution of 10,000 seed MOFs and report the results in Table 1. By design, the SELFIES algorithm guarantees a perfect score for molecular validity and, as evidenced by the high uniqueness and novelty scores in Table 1, it is clear that the model is not simply replicating linkers from the training set. The internal diversity (IntDiv) and similarity to nearest neighbour (SNN) metrics also confirm the chemical diversity of the generated linkers, indicating that the model's outputs extend beyond minor modifications of existing structures. Taking the SNN metric for $c_p^{min}$ @ 1 K as an example, a value of 0.226 implies that the dreamed distributions exhibit low similarities to the training distribution. To corroborate this, we conducted an experiment where we randomly sampled (without replacement) 1000 linkers from our training set and compared them to the rest. The result, an SNN value of 0.45 ± 0.001 (with the standard deviation calculated over five iterations), confirms that the generated linkers are more dissimilar from the training manifold than would be observed by a random sampling of existing structures. This point is further illustrated in Fig. 6, where we show the dreamed distributions in 2D phase space relative to the training set, clearly demonstrating that the model is charting largely unexplored areas of the chemical phase space.

As we utilise pretrained weights and biases for deep dreaming, another question arises on whether the model is capable of generalising to structures far removed from the training set. We do observe greater prediction errors for MOFs composed of linkers that are highly dissimilar to those observed during training. However, by quantifying the uncertainty in our model predictions, we can gauge where reliable

generalisations are likely as we chart these unknown areas (see Supplementary Note 7).

## Multiobjective optimisation

For most applications, MOF design is guided by multiple target objectives or by the satisfaction of certain constraints. Continuing with the theme of MOFs tailored for carbon capture applications, there are several properties one might be interested in optimising simultaneously. For DAC, one design objective at the material level is to maximise $Q_{CO_2}$, given that many physisorbents exhibit poor uptakes at the required process conditions. At the same time, the MOF should be selective to $CO_2$ over other gaseous components in the air, such as $N_2$ and $H_2O$. This leads to a dual design objective: maximising both the $Q_{CO_2}$ and the selectivity of $CO_2$ over gases such as, for example, $N_2$ (i.e., $S_{CO_2/N_2}$).

While there are other multiobjective design settings to be explored for DAC (Supplementary Note 8), here we seek to optimise both $Q_{CO_2}$ and $S_{CO_2/N_2}$ simultaneously from a seed distribution of 1000 MOFs. For this, we train a single model to predict both properties and invert the architecture for deep dreaming. Similar to the single objective case studies, Fig. 7a, b reveals shifts in the property distributions towards higher $Q_{CO_2}$ and $S_{CO_2/N_2}$ values. However, these shifts are less pronounced than those observed for $c_p$ (Fig. 5a) or, for example, the bandgap (Supplementary Fig. S5). We expect this arises for two related reasons: first, the mapping between MOF strings and the adsorption properties is less accurate with the sample size used to

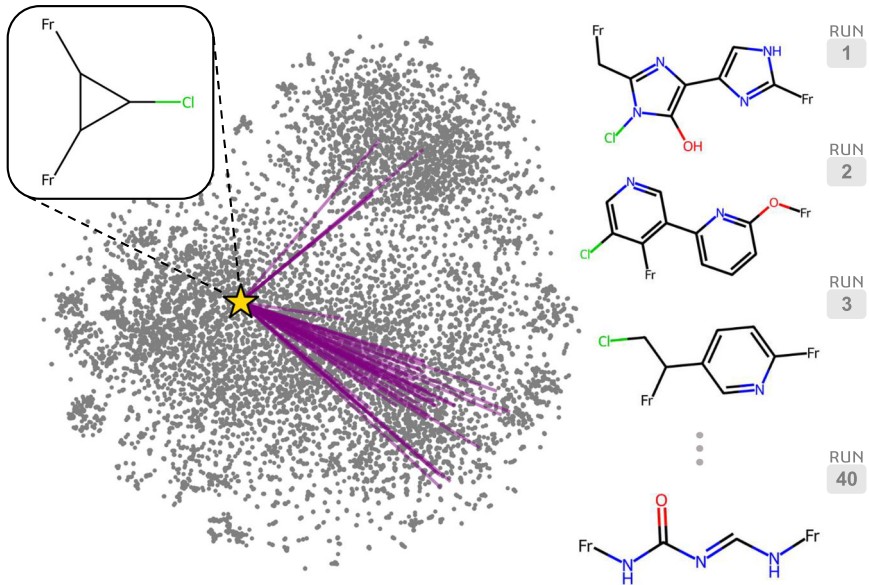

**Fig. 4 | Repeated trials of an edge secondary building unit (SBU) optimisation.** The same seed edge SBU (represented by the star point) initialised with different random noise 40 times, will produce different optimised structures due to the stochastic nature of the deep dreaming approach. We show where the optimised linkers lie within the chemical phase space relative to the training set (represented by grey points) using a t-SNE[68] projection of their Morgan fingerprints[69], computed with a radius of 2 and 1024 bits using RDKit[70]. The purple lines connect the seed structure and the final optimised structures. A few examples of the decoded linkers across several runs are shown on the right. Source data are provided as a Source Data file.

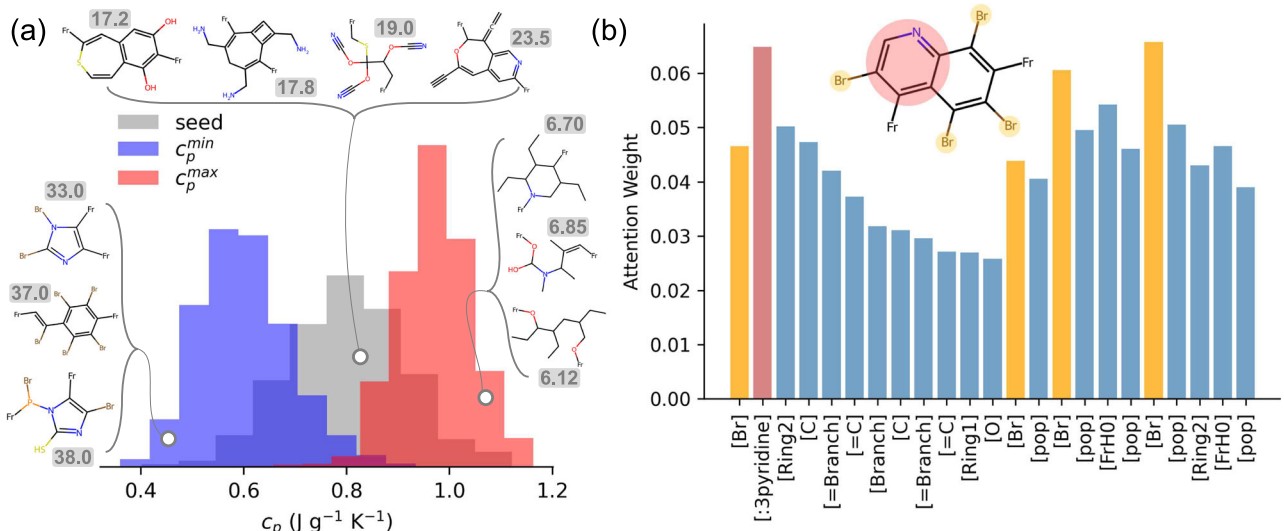

**Fig. 5 | Shifting specific heat capacity ($c_p$) property distributions. a** From a seed distribution of 1000 MOFs, the $c_p$ is either minimised ($c_p^{min}$) or maximised ($c_p^{max}$). Examples of linkers found in each of the three distributions are shown, with the average atomic mass of the metal-organic framework (MOF) constructed using these linkers given next to each structure. Note that the $c_p$ values of the dreamed distributions are computed using the machine learning model of Moosavi et al.[27] for the unrelaxed crystal structures. **b** Attention weights for the Group SELFIES tokens of an edge secondary building unit (as given by the molecular scheme shown) sampled from a MOF with $c_p = 0.48$. The pyridine substructure of the linker is highlighted in red, while the bromine functional groups are highlighted in orange. Source data are provided as a Source Data file.

train our model. Second, we do not posit any 3D coordinate information in our MOF strings. Given that adsorption properties are highly dependent on the spatial configuration of the atoms–evidenced by the $Q_{CO_2}$ and $S_{CO_2/N_2}$ distributions of the relaxed crystal structures in Supplementary Note 5–it is not surprising that this choice of representation is less effective at optimising these properties. Recent work by Alampara et al.[45] suggests that including this information may not necessarily enhance performance, as text-based approaches struggle to utilise this data effectively. Instead, they tend to emphasise local

chemical environments while neglecting global structural features, leading to challenges in capturing long-range interactions or periodicity. This reflects an inherent limitation of language models in application to material design. Nonetheless, the property shifts we observe here are still noteworthy and, similar to the $c_p$ case study, they are facilitated by designing MOF linkers with specific chemical groups identified by the model as significant contributors to the enhancement of $Q_{CO_2}$ and $S_{CO_2/N_2}$. Figure 7c highlights that certain Group SELFIES tokens, such as [benzenediol], [pyrimidine], and [imidazole],

**Table 1 | MOSES metrics for $c_p$ distribution optimisations**

| | Uniqueness | Novelty | IntDiv | SNN |
|---|---|---|---|---|
| $c_p^{min}$ @ 1 K | 0.993 | 0.995 | 0.894 | 0.226 |
| $c_p^{max}$ @ 1 K | 0.999 | 0.998 | 0.823 | 0.253 |
| $c_p^{min}$ @ 10 K | 0.994 | 0.999 | 0.901 | 0.219 |
| $c_p^{max}$ @ 10 K | 1.000 | 0.999 | 0.826 | 0.252 |

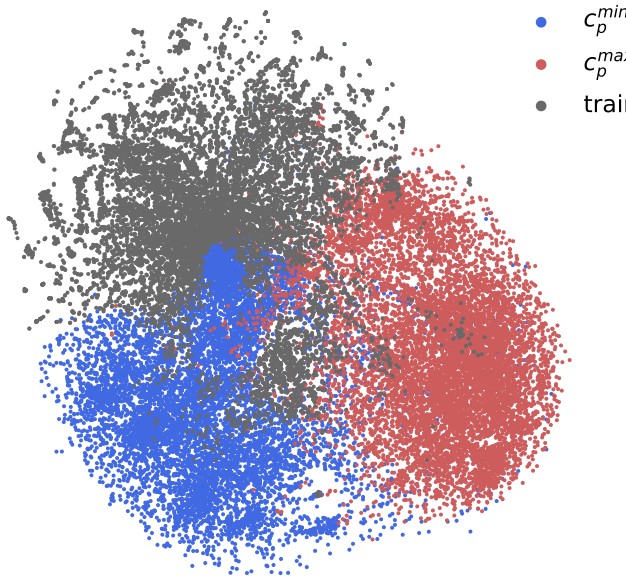

- $c_p^{min}$ (blue)
- $c_p^{max}$ (red)
- train (grey)

**Fig. 6 | t-SNE projection of Morgan fingerprints for linker molecules generated via deep dreaming.** Red and blue points represent metal-organic frameworks (MOFs) whose linkers were optimised to minimise ($c_p^{min}$) and maximise ($c_p^{max}$) specific heat capacity, respectively. Grey points denote the original training distribution for reference. 10,000 seed MOFs are used to generate the dreamed distributions. Source data are provided as a Source Data file.

are considered more important than, e.g., [alkene] and [alkyne] for this design task. That is, chemical substructures with more electrophilic character have greater learned attention weights, and so these Group tokens emerge more often in the dreaming process. Similarly, in Fig. 7d, we see that [P], [Cl], and [S] functional groups are deemed crucial for enhancing $CO_2$ adsorption in MOFs, as evidenced by their elevated incidence rates in the dreamed distribution compared to the seed distribution (Fig. 7e). Evidently, the model's understanding of the structure-property relationship, visualised via attention score distributions, is being exploited to reverse-engineer targeted MOF populations.

## Discussion

Deep dreaming is just one of several emerging models in the field of inverse MOF design, each varying in their core methodologies, data requirements for training and validation, and the realism of their predictions. This diversity prompts a practical question: which model should be used, and how can their performance be compared? Answering this is complicated by the challenge of defining robust, consistent performance metrics that capture the nuances of materials design. For instance, where generative models compete to optimise specific performance metrics, designs may excel according to these metrics yet fail to reflect the complexities of practical materials[20,46]. Or, the metrics themselves may not be sufficiently expressive to differentiate the performance of one generative model over another. These issues have been highlighted by established benchmarks in areas like

drug molecule design, demonstrating the need for more robust metrics that align with real-world applications[47].

These challenges become even more pronounced in the realm of MOFs, where a consistent benchmarking framework is notably absent. Currently, performance comparisons are only feasible for a selection of metrics. To contextualise our contribution within the broader field of inverse MOF design, in Supplementary Note 9 we provide a review of other generative models, including variational autoencoders (e.g., Sm-VAE[13]), diffusion-based approaches (e.g., GHP-MOFassemble[14]), and reinforcement learning transformers[5]. These models differ in their choice of representations (relying on latent space versus real space representations), in their design scope (from full MOF generation to linker-only optimisation), and in their training requirements (the amount of labelled/unlabelled data points needed for learning). We then compare the models in terms of decoding validity—which determines whether a MOF CIF file is correctly constructed from its latent or encoded representations—and the uniqueness of generated linkers. We reflect on the advantages and disadvantages of the available models, offering practical guidance for their selection. For example, deep dreaming achieves high validity (~95.5%) and uniqueness (~99.4%) using relatively little training data, particularly when compared to models such as Sm-VAE. These advantages, however, come at the cost of a narrower design scope: deep dreaming is limited to linker-only optimisation, whereas Sm-VAE supports full MOF structure optimisation, including topologies and metal nodes. Ultimately, the suitability of a given model depends on the specific goals and constraints of the practitioner, whether the priority lies in interpretability and data efficiency or in broader structural exploration.

That said, we must emphasise here that, while decoding validity and linker uniqueness offer some basis for comparison, these metrics still do not capture important practical considerations, such as the synthesisability of MOFs generated in silico. But, exactly defining what constitutes a 'synthesisable' MOF is still an open-ended and important problem[48]. While methods exist to indicate the expected difficulty of synthesising organic molecules—such as the synthetic accessibility (SA)[49] and synthetic complexity (SC)[50] scores (see Table 2 in the 'Methods' section)—they are somewhat limited when applied to MOFs. In some cases, reliance on these metrics may lead to misleading insights regarding the practicality of realising hypothetical MOFs. For example, organic molecules with SA scores > 6 are generally considered difficult to synthesise[49,51], and so linkers may be considered synthetically viable below this threshold[5]. According to this criterion, almost all the $c_p^{max}$ optimised linkers from Fig. 5a would be deemed feasible, as indicated by the SA score distribution in Fig. 8a. However, experts in MOF synthesis would likely have reservations about the stability of MOFs created with some of the 'synthetically accessible' molecules. This indicates that the SA score may not necessarily reflect the suitability of an organic linker. Conversely, the distribution of SC scores among the same set of optimised structures correlates much better with the 'linker-like' character of organic molecules. For instance, linkers with SC scores > 4 tend to be very large, highly asymmetric molecules with connection points oriented in such a way that the MOF structure would likely collapse upon activation. Linkers with SC scores < 4, however, have far more realistic features (such as fewer bulky side chains, reasonably oriented connection points, and lower molecular weights). These observations prompted us to explore a constrained $c_p^{max}$ optimisation, where the SC score is incorporated into an additional loss function, such that molecules with SC > 4 are heavily penalised and lower SC scores are encouraged. In Fig. 8b, we show that by constraining the dreaming process in this way, we can achieve a similar distribution of $c_p^{max}$ optimised MOFs while avoiding the formation of synthetically complex linkers.

It is clear from this case study that the SC score can be used to mitigate some of the linker feasibility concerns, yet it still remains a proxy to more rigorous definitions of MOF synthesisability. To compliment generative modelling outcomes, a comprehensive metric that

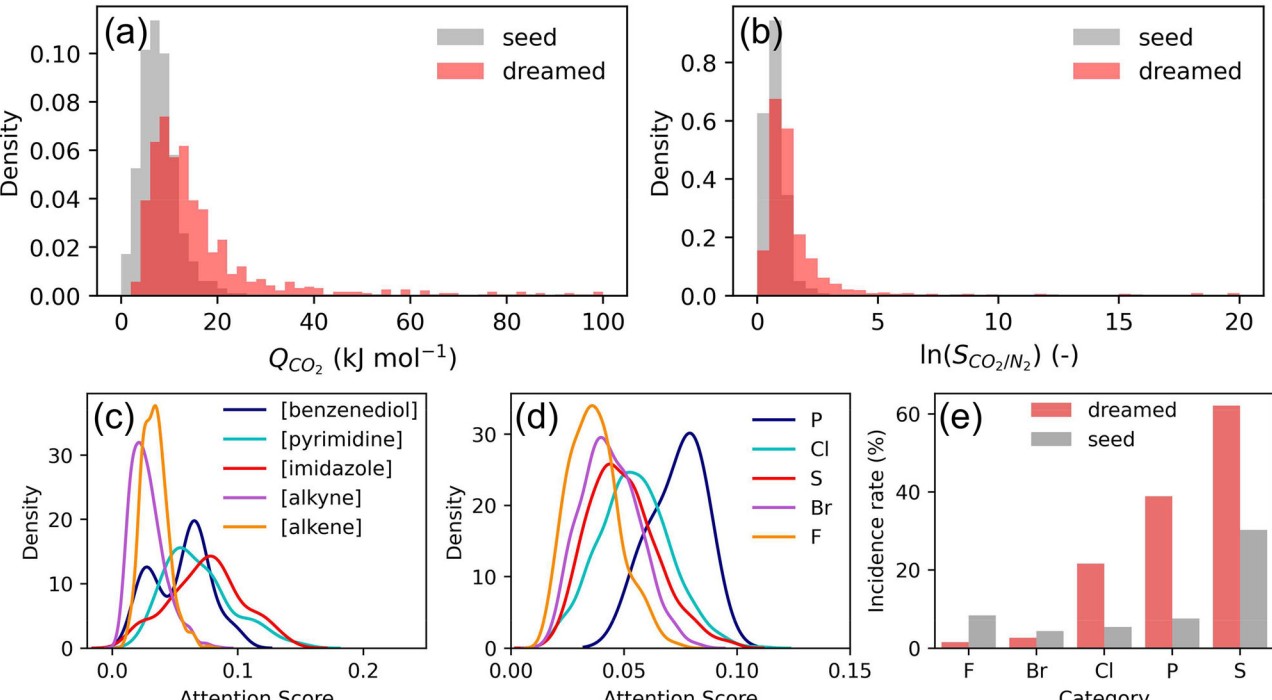

**Fig. 7 | Multiobjective optimisation of the heat of CO₂ adsorption at infinite dilution ($Q_{CO_2}$) and CO₂/N₂ Henry selectivity ($S_{CO_2/N_2}$).** From a seed distribution of 1000 MOFs, **a**, **b** show the dreamed distributions obtained from maximising $Q_{CO_2}$ and $S_{CO_2/N_2}$ simultaneously. Note that these distributions show the recomputed 'ground truth' values obtained by molecular simulation using unrelaxed crystal structures. **c**, **d** The distribution of attention scores learned by the model during training for select Group tokens and functional groups (see Supplementary Fig. S1 for the chemical structure of each Group token). **e** The incidence rate (% of linkers within the population that contains a certain chemical substructure) of the seed and dreamed distributions. Source data are provided as a Source Data file.

evaluates the entire MOF structure, such as, for example, the crystal free energy[52], would likely be necessary to encourage the experimental validation of simulation-guided MOF discoveries. Such a metric could be integrated as an additional output for direct multiobjective optimisation of MOF synthesisability alongside other KPIs, offering a more holistic approach than simply restricting focus to the synthesisability of individual linkers. Nonetheless, it is important to recognise that this framework might still inadequately address the risks associated with purely theoretical predictions, such as the thermodynamic stability or safety requirements involved in linker synthesis. Feasibility in these cases often relies on expert chemical intuition rather than forming an integral part of the property-driven optimisation, which can limit the practical impact of generative modelling results. Evidently, more work is needed to arrive at a set of indicators that effectively probe the performance of generative models suitable for downstream MOF applications.

In summary, our ML model shows promise for designing new MOFs with targeted properties. The picture that emerges from our study is as follows: (1) for properties that depend mostly on the chemistry (such as the $c_p$), deep dreaming can produce pronounced shifts in the property distributions. For structural properties that are relatively easy to correlate (such as the VF), the deep dreaming approach also performs reasonably well. For more complex MOF properties that are sensitive to the configuration of atoms in 3D space (such as the heat of adsorption at infinite dilution), notable shifts in the property distributions are observed. However, textual representations of MOFs alone are less effective for optimising these properties within the deep dreaming approach; (2) a cornerstone of our model is its emphasis on interpretability, which is provided by attention score distributions learned during training, and further through inspection of the transmutation pathways charted by our model when tasked with a particular design objective; and (3) more work is required to guide property-driven optimisation of MOFs towards realisable structures, potentially by incorporating safety and synthesisability metrics as

additional outputs in the multiobjective deep dreaming setting. Our results show that, indeed, a model's learned representations can be leveraged to reverse-engineer new MOF structures, consolidating the tasks of structure optimisation and property prediction within a single unified framework. We anticipate that future developments in the Group SELFIES algorithm[37] and refinements in our methodology will enable more targeted optimisation strategies, such as functional group substitution, providing deeper insights into the MOF structure-property relationship at an early stage.

## Methods

### Textual representation of MOFs

Similar to the MOF identification scheme (MOFid) of Bucior et al.[36], we use a string representation that captures the chemistry of MOF building blocks and some of the structural information through topology encoding. However, our representation differs from MOFid in two notable ways: (1) we consider MOF building blocks as secondary building units (SBUs)—which are defined by their connection points to other SBUs within a topological net—while MOFid's chemistry-focused syntax retains organic linkers as discrete building blocks, including any carboxylate groups[36]; (2) the chemistry of MOF building blocks is described here using SELFIES[34,37] rather than SMILES[53,54]. Specifically, Group SELFIES[37] is used to represent the edge SBUs (i.e., the linkers) while SELFIES is used for the node SBUs. This choice is motivated by the observation that chemical moieties important for MOF linker design, such as functional groups or aromatic rings, are rarely preserved when the SELFIES representation of edge SBUs is perturbed. However, these moieties can be explicitly encoded (and therefore more frequently preserved) by leveraging group tokens for entire chemical substructures, which is an advantage offered by Group SELFIES (see Supplementary Note 1).

While we use SELFIES as the foundation of our MOF string representation, it is important to note that other methods, such as

**Table 2 | Metrics for linker evaluation**

| Metric | Meaning |
|---|---|
| Novelty | Fraction of generated molecules that are not present in the training set. Limits are [0, 1]. |
| Uniqueness | Fraction of unique molecules generated. Limits are [0, 1]. |
| Internal diversity (IntDiv) | Assesses how dissimilar a linker is to the rest of the population. This metric effectively quantifies the chemical diversity within the generated set of linkers. A higher value corresponds to higher diversity in the generated set. Limits are [0, 1]. |
| Similarity to nearest neighbour (SNN) | Measures the average Tanimoto similarity between fingerprints of a generated linker and its nearest neighbour from the training set. If this value is low, then the generated linker is far from the manifold of the training set. Limits are [0, 1]. |
| Tanimoto similarity | Also known as the Jaccard index, this metric is a measure of the similarity between two linker molecules represented by binary fingerprints. |
| Synthetic accessibility (SA) score | A score which estimates the ease of synthesis for drug-like molecules by combining fragment contributions from molecular substructures with a complexity penalty that penalises, for example, complex ring systems and/or molecules with many stereo centres. Limits are [1, 10], where 1 means easy to make and 10 means very difficult to make. |
| Synthetic complexity (SC) score | A score calculated using a neural network trained on 12 million reactions from the Reaxys database that correlates with the expected number of reaction steps required to produce a molecule. Limits are [1, 5], where 1 means easy (simple) to synthesise and 5 means very difficult (complex) to synthesise. |

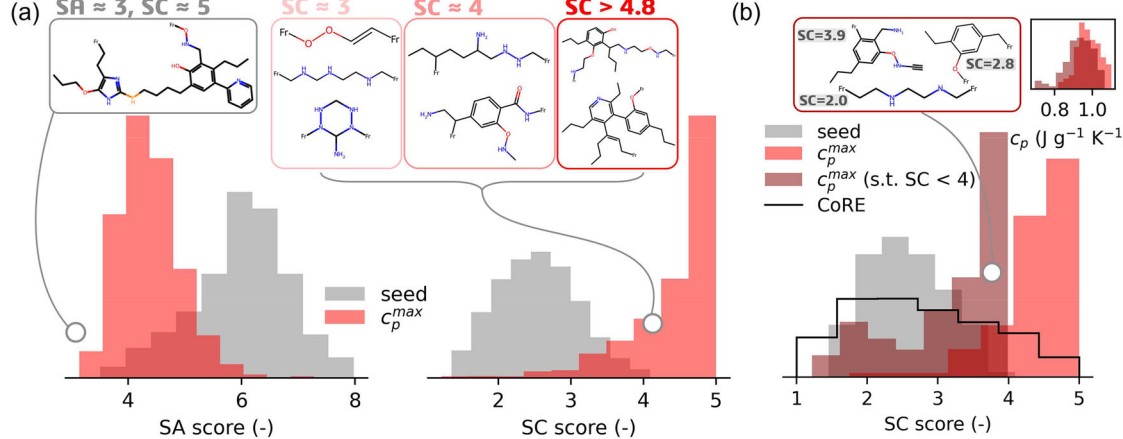

**Fig. 8 | Synthetic accessibility (SA) score vs synthetic complexity (SC) score distributions for metal-organic frameworks (MOFs) optimised for maximum specific heat capacity ($c_p^{max}$). a** On the left, the SA scores for the dreamed distribution are shifted left of the seed distribution. However, linkers with low SA scores can still be synthetically complex, as shown by the sampled molecule with an SA score = 3. On the right, the same pool of optimised linkers shows SC scores that are systematically shifted to higher values than the seed distribution, reflecting this complexity. Molecules with SC score < 4 show reasonable 'linker-like' characteristics, while those above this threshold are likely too complex/unstable for MOF design. **b** We perform a constrained (s.t. SC score < 4) and unconstrained $c_p^{max}$ optimisation from a seed of 100 MOFs. We achieve an SC score distribution in the constrained case study that more closely reflects the distribution of experimental organic linkers from the CoRE MOF[8] database, resulting in more realistic structures generated during dreaming, while still achieving similar $c_p^{max}$ optimised distributions (as shown in the top right inset). Source data are provided as a Source Data file.

SMILES[53] and DeepSMILES[55], may also be used to represent molecules in ML applications, as discussed by Krenn et al.[47]. The oft-cited advantage of SMILES is its human readability. However, an important limitation is its tendency to produce strings that do not map to valid molecular graphs, which can be attributed to the fragile grammar. While it is yet to be established whether, in general, SELFIES or SMILES strings yield better ML learning outcomes, we find that SMILES tends to perform better in the forward process of our model. Nevertheless, our focus is on the reverse process, where the syntactic and semantic robustness of SELFIES notation becomes crucial (see Supplementary Note 10).

### Dataset curation
As our deep dreaming framework operates in a data-driven setting, we require a large collection of MOF structures and their corresponding KPIs. To that end, we constructed a diverse database of MOFs using pormake[16]. We first expanded the relatively limited number of edge building blocks available in pormake with additional organic linkers from the augmented edge dataset of Yao et al.[13], which encodes ~300,000 linker molecules as SMILES strings. From

this extended pool of edge SBUs, we construct MOFs with unique organic linkers to ensure sufficient chemical diversity in the training dataset. The diversity during construction was limited to frequently occurring topologies with one node/edge building block per MOF to align with experimental synthesis feasibility[16] (a list of the topologies which satisfy our criteria is provided in Supplementary Table S4). We further discarded any MOFs with cell lengths greater than 60 Å or whose number of atoms exceeded 1500 to minimise the computational cost associated with material property predictions. For MOFs that undergo structural relaxation, we use the Universal Force Field (UFF)[56] as implemented in the Forcite Module of Materials Studio 2019[57]. While our ML model is currently tailored to optimise MOFs featuring a single unique edge SBU, we anticipate that future modifications to the architectural design will accommodate MOFs with multiple unique edge SBUs.

### Deep dreaming model architecture
The deep dreaming model is composed of three primary components: (1) LSTM cells; (2) softmax self-attention mechanisms; and (3) MLP neural networks.

LSTMs are a variation of recurrent neural networks that introduce specialised units, known as LSTM cells, to process data sequentially. At a given timestep, each LSTM cell processes a single element of the input sequence $x_t$, in conjunction with the network's preceding hidden state $h_{t-1}$. This structure equips LSTMs with a distinctive advantage in capturing long-range relations within sequences[58]. The LSTM cells, composed of a forget gate, input gate, and output gate, collectively decide what information to retain or discard as the sequence progresses. This architectural design allows LSTMs to excel in settings where understanding extended context is crucial[59], such as in correlating MOF strings with their material KPIs. We chose this architecture over simpler models, like a fully connected feed-forward neural network (as in the work of Shen et al.[26]), as we found they were not capable of correlating MOF strings to their property values with sufficient accuracy. We also ruled out more advanced models like transformers from consideration. Unlike our chosen architecture, transformers process input sequences using integer vectors rather than noisy one-hot encoded floating point vectors. This restriction limits their ability to modify input sequences based on gradient information.

We do, however, incorporate self-attention into our model architecture, a key feature inspired by the transformer model. Each hidden state output by the LSTM is transformed individually by a linear layer, which projects the hidden state vector to a scalar value:

$$a_t = W \cdot h_t + b \tag{1}$$

Where $a_t$ is the raw attention score for timestep $t$, $h_t$ is the hidden state at timestep $t$, $W$ is the weight matrix, and $b$ is the bias (where $W$ and $b$ are parameters of the linear layer). The raw attention scores across all timesteps are normalised using a softmax function to convert them into a probability distribution, indicating the relative importance of each timestep's hidden state. This can be expressed as:

$$\alpha_t = \frac{\exp(a_t)}{\sum_{j=1}^{T} \exp(a_j)} \tag{2}$$

Where $\alpha_t$ is the softmax-normalised attention weight for timestep $t$ and $T$ is the total number of timesteps. The context vector is calculated as a weighted sum of all LSTM hidden states, with weights determined by the attention scores. This operation effectively allows the model to 'focus' more on certain parts of the sequence deemed more relevant by the attention mechanism. In essence, the context vector $c$ summarises information across the entire sequence weighted by its computed importance:

$$c = \sum_{t=1}^{T} \alpha_t \cdot h_t \tag{3}$$

By leveraging this mechanism, our architecture not only retains the LSTM's capability to process sequences as differentiable entities but also gains from the attention-driven focus of transformers.

### Training details
After some rudimentary preliminary testing, we settled on a consistent set of hyperparameters for each of the models considered in our study, which are detailed in Table 3. We train each model architecture using the Adam optimiser and a learning rate of 0.001 with 10,000/1000/1000 train, validation, and test data points, respectively. During training, the model learns a partial mapping between the input and target values before any noise is added to the one-hot encoded edge SBUs. Noise is then gradually introduced to the zero elements by sampling from a uniform distribution between 0 and $k$, where $k$ is increased from 0 to 0.6 in increments of 0.1. This injection period helps the model to learn the underlying patterns in the presence of varying levels of noise. Once $k = 0.6$, the model is trained until the prediction losses on the validation

dataset fail to improve over 10 epochs. The choice of $k$'s upper limit at 0.6 represents a strategic balance: it introduces enough noise to ensure meaningful gradient updates to the input feature vector during optimisation, while maintaining a high level of predictive accuracy compared to the noise-free representation. However, while adding noise is necessary to backpropagate gradient information in the reverse process, this fuzzy representation of the linkers does still lead to a less accurate mapping between MOF strings and their respective KPIs (see Supplementary Fig. S3). Therefore, we couple the deep dreaming model—trained on noisy MOF strings—with a predictor model of the same architecture trained on the noise-free representations. The driver for molecular optimisation is provided by the dreaming model, while the final prediction from the reverse process is obtained from the predictor.

### Deep dreaming details
Deep dreaming is achieved in the same way the model is trained, except the MOF linker encoding is what is being updated in the reverse process, rather than the weights and biases of the model. We use the same noise levels ($k = 0.6$) to encode seed structures for deep dreaming. Furthermore, for all of our experiments, we use the Adam optimiser with a learning rate of 0.001. Dreaming is terminated if the predicted property value is within 10% of the target value or if the number of dreaming epochs exceeds 5000.

### MOF property determination
For both our training dataset and for MOFs reconstructed from their optimised string representations (using pormake), we compute several properties. The bandgap is determined using the average prediction from an ensemble of crystal graph convolutional neural networks (CGCNNs)[60] trained on data available from the QMOF database[2]. Ten thousand MOFs from the hybrid-level HSE06-D3(BJ) functional dataset were split into train/validation/test partitions using ratios of 8/1/1, respectively. Using an optimal CGCNN architecture (i.e., 5 convolutional layers, 64 hidden atom features in each convolutional layer, 1 fully connected hidden layer after pooling, and 128 hidden features after pooling, as determined in the original work of Rosen et al.[2]), 5 separate models are trained with different data partitions for up to 200 epochs using the Adam optimiser with a batch size of 256 and a learning rate of 0.01. The CGCNN ensemble predicts MOF bandgaps with an MAE of 0.307 ± 0.006 eV, which is consistent with the accuracy of other CGCNNs reported in the literature for this task[2,39].

To generate training data for the $c_p$, we use the gradient-boosted decision tree model of Moosavi et al.[27], which predicts out-of-sample testing data with an MAE of 0.02 J g$^{-1}$ K$^{-1}$. $c_p$ predictions are obtained at 300K.

The $Q_{CO_2}$ and Henry coefficients of $CO_2$ and $N_2$ adsorption were determined by Widom's particle insertion algorithm at 300 K using the RASPA[61] simulation package. Five thousand configurational-biased insertions of $CO_2$ and $N_2$ were used for each structure. Interaction energies between non-bonded atoms were computed using the Lennard–Jones (LJ) plus Coulomb potential[62], taking LJ parameters for framework atoms and adsorbed molecules from the UFF and TraPPE[63] force field, respectively. For cross-interactions between unlike atoms, the LJ parameters were approximated using Lorentz–Berthelot mixing rules. A simulation supercell with all three dimensions greater than twice the LJ potential cut-off distance of 12 Å was used to satisfy the minimum image convention. To assign partial atomic charges, the extended charge equilibration (EQeq) method[64] was used. While we have shown previously that this method for partial charge assignment is unsuitable for process-level screening applications[62,65], we restrict ourselves to a material-level analysis in this work as a simple demonstration of the deep dreaming approach. For this objective, EQeq charges are sufficient. The Ewald sum technique was applied to compute the long-range electrostatic interactions with a relative precision of $10^{-6}$. To

**Table 3 | Deep dreaming model hyperparameters (PyTorch v2.2.1 LSTM module)[a]**

| | |
|---|---|
| `input_size` (LSTM cell for edge encoding) | 39 |
| `input_size` (LSTM cell for node and topology encoding)[b] | 10 |
| `hidden_size` | 50 |
| `num_layers` | 1 |
| `dropout` | 0.1 |
| `bias` | True |
| `batch_first` | True |
| `bidirectional` | False |
| `proj_size` | 0 |

[a]Softmax attention is applied to the outputs of the LSTM cells, which are then concatenated (`hidden_size`×2) and passed through a multilayer perceptron (`input_size`=`hidden_size`×2, `num_layers`=3, `output_size`=$N$, where $N$ = number of KPIs we are trying to predict).
[b]Input size is obtained by embedding the node and topology encoding (of max length 103) using a vector of length 10.

compute the Henry $CO_2/N_2$ selectivity, $S_{CO_2/N_2}$, we use Eq. (4):

$$S_{CO_2/N_2} = \frac{K_H^{CO_2}}{K_H^{N_2}} \quad (4)$$

Finally, the GSA and VF were calculated using Zeo++v0.3[66] with a probe radius of 1.86 Å and high accuracy settings. We consider both the accessible and the inaccessible contributions for each of these properties.

### Linker evaluation metrics

The quality of generated linker molecules is evaluated using several metrics, mainly from the MOSES[44] framework, as described in Table 2. Aside from the uniqueness and novelty scores, we also compute the IntDiv and SNN scores[44].

The IntDiv scores were computed by Eq. (5) using the Tanimoto similarity, $T_s(m_1, m_2)$, between Morgan fingerprints among all pairs of AI-generated linkers ($m_1$ and $m_2$). As $\sum_{m_1, m_2 \in G} T_s(m_1, m_2)^2 \to 0$, then IntDiv($G$) → 1, indicating a population ($G$) with high diversity.

$$\text{IntDiv}(G) = 1 - \sqrt{\frac{1}{|G|^2} \sum_{m_1, m_2 \in G} T_s(m_1, m_2)^2} \quad (5)$$

The SNN metric, given by Eq. (6), computes the average Tanimoto similarity between Morgan fingerprints of an AI-generated linker ($m_G$) and its nearest neighbour in the reference set ($R$). In our case, the reference set is taken to be the training set.

$$\text{SNN}(G, R) = \frac{1}{|G|} \sum_{m_G \in G} \max_{m_R \in R} T_s(m_G, m_R) \quad (6)$$

### Reporting summary

Further information on research design is available in the Nature Portfolio Reporting Summary linked to this article.

### Data availability

The datasets and trained models, dreaming results, and associated CIF files, are available on GitHub (https://github.com/SarkisovTeam/dreaming4MOFs.git) and on Zenodo[67]. Source data are provided with this paper.

### Code availability

All code created in this work is available on GitHub (https://github.com/SarkisovTeam/dreaming4MOFs.git) and on Zenodo[67].

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

## Acknowledgements

The authors would like to thank Dr Joseph Manning for his comments and feedback on the manuscript. The authors would also like to acknowledge the assistance given by Research IT and the use of the Computational Shared Facility at the University of Manchester.

## Author contributions

C.C. contributed to the conceptualisation of this work, developed the machine learning model, led the computational studies, led the data analysis, and wrote the original draft. L.S. contributed to the conceptualisation of this work, contributed to the data analysis, led the supervision of the work, and wrote the original draft.

## Competing interests

The authors declare no competing interests.
