## [Transparent Peer Review file · Nature Communications]

Inverse design of metal-organic frameworks using deep dreaming approaches

Corresponding Author: Mr Conor Cleeton

Version 1:

Reviewer comments:

Reviewer #1

(Remarks to the Author)

The paper proposes a new method for the design of metal-organic frameworks (MOFs) using deep dreaming. The major differentiator for the proposed deep dreaming approach, which was inspired by prior work in organic molecule design, lies in using gradient-based optimization to traverse the molecular space directly after fixing the weights of the deep learning model. The paper first introduces and motivates the design of MOFs for diverse applications, followed by a description of generative models for MOFs and drug-like molecules. Next, the paper describes the motivation for their proposed deep dreaming method based on prior work in organic molecules and outlines the proposed extensions to apply deep dreaming to MOFs.

In the subsequent section, the paper details the deep dreaming approach in greater detail. The primary difference in deep dreaming compared to other generative models is that gradient based optimization is used to adjust the input while keeping the weights of the model architecture fixed. Next, the paper describes the proposed design setting for MOFs into three separate parts (edge SBU, node SBU, topology representation) and their respective data structures. Following the description of the method, the paper outlines a set of MOF design experiments starting with optimizing individual MOF structures based on a single objective. The deep dreaming approach enables the authors to analyze the transmutation pathways of the proposed molecules in greater detail and see how the model modifies each other the proposed molecules. The authors perform this optimization and analysis for a couple of cases and observe successful design and shifting property distributions.

Following the discussions of the single property optimization, the paper presents a case of multi-objective optimization using deep dreaming and describes some of the shortcomings of the current text-based design approach. The paper then concludes with a discussion of the results and future work that can address shortcomings of the current state-of-the-art. Most of the discussion of future work focuses on being able to model more complex properties more accurately.

Overall, the paper provides a novel method for MOF design with compelling case studies and results. The proposed method has the advantage of being able to interpret the transmutation space, which can give chemists additional confidence in the suggestions made by the generative technique. Moreover, the paper provides relevant methodological descriptions and details along with pertinent analysis of the results. While the current draft is quite promising, it could be further improved by:

1. Exploring whether using pretrained property prediction or generative models have an effect on the goodness of the representations for the proposed MOF design approach. One particular relevant work is [1] which applies a pertaining + RL approach for molecular drug design based on docking scores. Since the representations are a key part of the proposed method it would be good additional data on this that explore representations from pretrained models directly, as well as representations from pretrained+finetuned models.
2. Adding details on how to perform automated interpretation of the proposed deep dreaming pathways. Since this is the most distinguishing feature of the proposed method, it would be good for the community to be able to take advantage of that. This can also serve as an example in the code repo.
3. Additional discussion of representations for molecular design [2], as well as the advantages and disadvantages of using text-based representations since the paper mentions that not including geometry leads to modeling shortcomings of certain

properties. I would also encourage the authors to look at additional work that has exposed shortcomings of text-based representation in materials design [3] that can influence future work.

[1] Ghugare, Raj, et al. "Searching for High-Value Molecules Using Reinforcement Learning and Transformers." The Twelfth International Conference on Learning Representations.

[2] Krenn, Mario, et al. "SELFIES and the future of molecular string representations." Patterns 3.10 (2022).

[3] Alampara, Nawaf, Santiago Miret, and Kevin Maik Jablonka. "MatText: Do Language Models Need More than Text & Scale for Materials Modeling?." AI for Accelerated Materials Design-Vienna 2024.

(Remarks on code availability)

The repo is generally well-documented outlining relevant requirements and scripts. I would recommend providing the full dataset for MOF construction to replicate the experiments in the paper, as well as creating or pointing notebooks for the interpretability analysis.

Reviewer #2

(Remarks to the Author)

This manuscript discusses MOF generation strategy which integrates material property evaluation during the generation such that generated MOFs are only shifted towards certain properties of interest, instead of blindly generating large number of structures. This strategy is largely motivated by the potentially unlimited number of MOFs that can be generated in silico. Authors dub this strategy as "deep dreaming" but this strategy is simpler just inverse design of materials. Having said this, I think the manuscript is an interesting paper that could find potential utility in the ongoing effort in the field to systematically generate hypothetical MOFs, and I support the publication of the manuscript. There are a few comments that I would like the authors to address and I leave them below.

Comment 1: Is the developed ML model applicable only to MOFs with a single organic linker, or can it also be applied to MOFs containing two or three different linkers?

Comment 2: When optimizing the ligands, does the authors considered only ligands with two connection points? If not, it would be better to include examples of optimizing ligands with three or more connection points.

Comment 3: In optimizing MOF properties, where the ligand is optimized, does the model include structural optimization of the MOF? If so, what method was used for the optimization—DFT or MLP-based optimization? The property prediction results may vary depending on whether the MOF's structure has been optimized. If it is too cumbersome to include this as part of the manuscript, it would be good to add a brief discussion of potentially integrating this as part of the workflow.

Comment 4: As shown in the property predictions in Table S7, the model shows poor predictions for adsorption-related properties such as heat of adsorption and CO₂/N₂ selectivity. Does this mean that MOF optimization using this model cannot be applied to improving adsorption performance and is only applicable to texture properties? Is there a specific reason why the author did not include 3D coordinate information to improve the prediction of those?

Comment 5: It would be better for the author to include prediction results for other texture properties such as surface area, pore size, etc.

Comment 6: In the MOF optimization, does the model also consider ligand functionalization, such as functional group substitution? This is known to have a significant impact on improving adsorption performance.

(Remarks on code availability)

Reviewer #3

(Remarks to the Author)

In this manuscript, Cleaton, et al. utilizes inceptionism for inverse design of MOF linkers. They train a LSTM enhanced with attention between node and edge embeddings, together with MLP, on synthetic MOF data for this purpose. The paper, for most part, was written clearly, though there are several missing details (see comments). While the problem is important, I did not find the work comprehensive in terms of comparing its performance over existing baselines or in terms of validation of optimized molecules. Neither I found the technology leveraged is novel enough. Therefore, my recommendation is that the paper, as it is, is not suitable for publication in Nature Communications. I have listed my comments below, which I believe can be helpful for improving the work.

Comments:

(1) It is not very clear what is the advantage of the proposed method. Is it the ability to train on small number of samples? Is that a big constraint when the data is synthetic?

(2) Can the authors use the code in the available GitHub repo to train a Sm-VAE that uses the synthetic data used in this study for the semi-supervised part of the training. That way, it will be possible provide a vis-a-vis comparison on property-optimized molecules using the proposed method and Sm-VAE.

(3) Can the authors provide an equation that describes the full training loss? From the description, there seems to be a property prediction loss plus a penalty on invalid linkers; however, no further details have been provided.

(4) The authors should consider a figure to illustrate the forward learning and inverse training, as in the original Shen, et al. paper to help reader understand the generative model. Current writing leaves it ambiguous.

(5) As the results suggest that generated molecules are highly novel (low SNN) with respect to training distribution, how do we know that the predictive model generalize reliably on this novel distribution?

(Remarks on code availability)

Version 2:

Reviewer comments:

Reviewer #1

(Remarks to the Author)

I thank the authors for their response and edits to the manuscript. I think the manuscript clarified much of the initial feedback. I am in favor of supporting the manuscript, assuming the following updates can be made:

1. Adding additional references on applying reinforcement learning type techniques to materials design, including MOF (ref. 8 is already included), molecules [1], proteins [2], and crystal structures [3], to line 67 clarifying the procedure: "A similar principle applies to reinforcement learning: a pretrained structure generator (acting as the agent) learns to create new structures based on a reward system provided by a property predictor (serving as the environment), with the objective of maximizing the reward."

2. Adding a discussion on the importance and difficulties of doing proper benchmark for ML-based materials design. This could be done in the Discussion section where in line 401 the authors mention the difficulty of benchmarking. One common problem with benchmarking of ML-based generation is the metrics used for benchmarking are underspecified, which limits their utility and can lead to design that maximize the metrics, but are not useful for downstream applications. Some useful references include [1],[4],[5].

[1] Ghugare, Raj, et al. "Searching for High-Value Molecules Using Reinforcement Learning and Transformers." The Twelfth International Conference on Learning Representations.

[2] Lutz, Isaac D., et al. "Top-down design of protein architectures with reinforcement learning." Science 380.6642 (2023): 266-273.

[3] Govindarajan, Prashant, et al. "Learning conditional policies for crystal design using offline reinforcement learning." Digital Discovery 3.4 (2024): 769-785.

[4] Steshin, Simon. "Lo-hi: Practical ml drug discovery benchmark." Advances in Neural Information Processing Systems 36 (2023): 64526-64554.

[5] Jain, Moksh, et al. "Multi-objective gflownets." International conference on machine learning. PMLR, 2023.

(Remarks on code availability)

I appreciate the additions to the code repo.

Reviewer #2

(Remarks to the Author)

Authors have sufficiently answers my comments and the manuscript may be accepted for publication. I do not have further questions.

(Remarks on code availability)

Authors' Comments

We would like to take this opportunity to thank the reviewers for their useful comments and insights. We have carefully examined the reviewer's comments, and we believe all the issues raised have been addressed in this response document.

Throughout the following responses, we use double quoted italic text in green ("*example*") to indicate text taken from the originally submitted article and supplementary information, and bolded green text to indicate proposed changes for the revised article and supplementary information ("**modified example**"). Our response to reviewers' comments will be indicated by normal text coloured blue (example), while original comments by the reviewers will be indicated by normal text coloured black (example). Note that we use the following naming conventions to refer to figures in this response document: Figure X (figure in the revised manuscript); Figure SX (figure in the revised supplementary information); Figure RX (figure in this response document not included in the revised manuscript or supplementary information but is used to aid in the response to individual reviewer comments).

Reviewer #1 (Remarks to the Author):

Remark – The paper proposes a new method for the design of metal-organic frameworks (MOFs) using deep dreaming. The major differentiator for the proposed deep dreaming approach, which was inspired by prior work in organic molecule design, lies in using gradient-based optimization to traverse the molecular space directly after fixing the weights of the deep learning model. The paper first introduces and motivates the design of MOFs for diverse applications, followed by a description of generative models for MOFs and drug-like molecules. Next, the paper describes the motivation for their proposed deep dreaming method based on prior work in organic molecules and outlines the proposed extensions to apply deep dreaming to MOFs.

In the subsequent section, the paper details the deep dreaming approach in greater detail. The primary difference in deep dreaming compared to other generative models is that gradient based optimization is used to adjust the input while keeping the weights of the model architecture fixed. Next, the paper describes the proposed design setting for MOFs into three separate parts (edge SBU, node SBU, topology representation) and their respective data structures. Following the description of the method, the paper outlines a set of MOF design experiments starting with optimizing individual MOF structures based on a single objective. The deep dreaming approach enables the authors to analyze the transmutation pathways of the proposed molecules in greater detail and see how the model modifies each other the proposed molecules. The authors perform this optimization and analysis for a couple of cases and observe successful design and shifting property distributions.

Following the discussions of the single property optimization, the paper presents a case of multi-objective optimization using deep dreaming and describes some of the shortcomings of the current text-based design approach. The paper then concludes with a discussion of the results and future work that can address shortcomings of the current state-of-the-art. Most of the discussion of future work focuses on being able to model more complex properties more accurately.

Overall, the paper provides a novel method for MOF design with compelling case studies and results. The proposed method has the advantage of being able to interpret the transmutation space, which can give chemists additional confidence in the suggestions made by the generative technique. Moreover, the paper provides relevant methodological descriptions and details along with pertinent analysis of the results.

Response – We thank the reviewer for their positive feedback on the manuscript, and for their suggestions on improving the work in the following comments. We hope that, in each response below, we have addressed the reviewer's questions and queries regarding our work.

Reviewer #1 (Comments):

Comment 1.1 – Exploring whether using pretrained property prediction or generative models have an effect on the goodness of the representations for the proposed MOF design approach. One particular relevant work is [1] which applies a pertaining + RL approach for molecular drug design based on docking scores. Since the representations are a key part of the proposed method it would be good additional data on this that explore representations from pretrained models directly, as well as representations from pretrained+finetuned models.

Response 1.1 – The reviewer raises an interesting point, and indeed would be an avenue worth pursuing in related works of MOF inverse design, such as the transformer + RL approach of Park et al. (2024). However, there are a few concepts worth clarifying here to outline why this is not aligned with the scope of the current manuscript.

Deep dreaming is not a traditional generative model: the study the reviewer cites is referring to a “traditional” sequence-generation task, whereby a language model architecture (i.e., VAE, RNN, transformer) undergoes self-supervised generative learning to sample molecules or materials that resemble an unlabelled training dataset (pretraining). Typically, at this stage, a large number of unlabelled samples are used (from hundreds of thousands to millions). The generative process can then be biased towards molecules or materials with desired functionalities by finetuning with a smaller number of labelled data points (usually around tens of thousands). In the paper cited by the reviewer, this finetuning occurs by reinforcement learning, whereby the agent (generative model) learns a policy for creating new structures from a reward that is returned from the environment (property generator), with the objective of maximising some reward. Deep dreaming is architecturally different from this traditional generative modelling approach. We allude to this point in Supplementary note 10 of the revised supporting information, where we provide a comparison between deep dreaming and other generative models for inverse MOF design:

“Deep dreaming differs from typical generative models. Our approach does not learn to unconditionally generate samples that follow a training distribution like, for example, the Sm-VAE or RL transformer models. Our model only performs conditional (targeted) generation, as we learn a mapping between structures and their properties first ($f(X) \rightarrow \hat{y}$) and then invert the architecture to obtain new MOFs ($f(\hat{y}) \rightarrow X$) with improved property values. To do this, we must always define a value (\hat{y}) and optimise (X) towards this value.”

Said another way, we utilise labelled data points to train a *regression* model, and then invert the architecture to perform the generative modelling task. Conceptually, this may be thought of as skipping directly to the finetuning phase. As a result, we avoid the pretraining stage, meaning that the design choices made during this phase do not bear relevance to our results.

Choice of textual representation: the second point we would like to raise is that we chose the SELFIES representation for our model due to its compatibility with the deep dreaming approach, where we modify the MOF linker representation directly in real chemical space using gradient-based algorithms. This works best with an optimisation landscape where every point is chemically valid – a condition that SELFIES, known for its semantic and syntactic robustness, uniquely fulfils. Our experiments comparing MOF linker representations using SMILES and DeepSMILES highlight this point. For example, while SMILES or DeepSMILES may perform well in regression tasks (as shown in Figure R1 below), they fall short in the deep dreaming process, which is our main focus. This is because the ML model trained with these representations often produce semantically or syntactically invalid molecular strings, making the deep dreaming process much less effective. By comparing the optimisation results of the same MOF structure using these three representations (illustrated in Figure R2), we demonstrate that SELFIES outperforms the alternatives due to the high occurrence of semantically or syntactically invalid molecular strings. We specifically opt for the Group SELFIES

variant to preserve important chemical substructures in MOF linkers, as pointed out in the *Methods* section of the manuscript:

“Specifically, Group SELFIES is used to represent the edge SBUs (i.e., the linkers) while SELFIES is used for the node SBUs. This choice is motivated by the observation that chemical moiety important for MOF linker design, such as functional groups or aromatic rings, are rarely preserved when the SELFIES representation of edge SBUs are perturbed. However, these moieties can be explicitly encoded (and therefore more frequently preserved) by leveraging group tokens for entire chemical substructures, which is an advantage offered by Group SELFIES (see Supplementary Note 1).” – Lines 487-494

Figure R1. Comparison of the c_p out-of-sample performance for a sequence-to-regression ML model trained on different molecular string representations of MOF linkers, based on Group SELFIES (left), SMILES (middle), and DeepSMILES (right) strings. We use 10,000, 1,000, and 1,000 MOFs for training, validation, and testing purposes in each case study, and apply noise to the zero-elements of the one-hot encodings by sampling from a uniform distribution between [0,0.6].

Figure R2. A single deep dreaming case study, whereby we attempt to optimise the c_p of a seed MOF towards higher values, using ML models trained on different molecular string representations to encode the linkers of MOFs. In the case of SMILES, most of the generated strings are syntactically incorrect (violate the SMILES vocabulary), while a smaller number of generated strings (3.9%) are syntactically correct but do not correspond to a chemically valid molecular graph (i.e., are semantically incorrect). DeepSMILES – developed to overcome syntactic limitations of SMILES strings – produce fewer syntactically incorrect strings, but still fail to overcome the semantic limitations within the deep dreaming framework. SELFIES strings, on the other hand, only produce valid molecules, 31.9% of which are valid linkers (i.e., have the correct number of connection points, represented as francium pseudoatoms).

Choice of model architecture: the final point we wish to discuss concerns the choice of our ML architecture, which emerged from a careful model development process. We require a model that can handle sequence-to-regression tasks. As noted in the paper cited by the reviewer, the common architectures for this purpose would include fully connected neural networks (FC), recurrent neural networks (RNN) and their derivatives such as LSTM and bi-directional LSTMs, and transformers. We ruled out the use of a transformer architecture early in the development due to its unsuitability for deep dreaming experiments, as detailed in Supplementary Note 3:

“We chose this architecture over more advanced models like Transformers due to the LSTM model’s ability to handle sequences as continuously differentiable inputs. This is achieved through the use of noisy one-hot encoded floating-point vectors, making it suitable for gradient-based deep dreaming experiments. Conversely, traditional Transformers tokenize input sequences with integer vectors, which precludes the possibility of altering the input sequence based on gradient information.”

In deciding between RNN-type models, such as LSTM networks, and more basic FC models, we initially assessed the performance of each architecture. The FC models underperformed in the forward process, where the task was to train the regression model to predict various MOF properties. For instance, using the FC model, the mean absolute error (MAE) for properties like void fraction (VF), gravitational surface area (GSA), and Henry’s CO₂ / N₂ selectivity (S_{CO_2/N_2}) were 0.11 [-], 855 [m²/g], and 25.78 [-], respectively (Figure R3 below). Juxtaposed with our regression results in Table S7 of the supplementary information for the same properties (MAE of 0.05 [-], 368 [m²/g], and 2.24 [-]), it is clear that the current model architecture outperforms FC by a considerable margin and, hence, performs better in the inverse design task. We further compared bi-directional LSTM models against standard LSTM models and found no difference in the performance. As such, we opted for the standard LSTM model to minimise model training time.

Figure R3. Parity plots of the deep dreaming model regression performance for VF, GSA, and S_{CO_2/N_2} MOF properties, based on the FC architecture and Group SELFIES MOF string representation.

Proposed modifications to the manuscript: we propose the following modification to the manuscript to clarify, early on, the difference between deep dreaming and traditional generative models:

“A recurring theme in these studies is the strategy of pairing a structure-generating module with a property-predicting module, where communication between the two is limited by a reward or loss function. For example, in VAEs, GANs, and diffusion models, a structure generator learns to sample molecules or materials that resemble an unlabelled dataset through extensive self-supervised pretraining using hundreds of thousands of data points. The generator is then directed by the property predictor — trained on a smaller subset of labelled data points — through an optimisation process that explores a latent chemical space. A similar principle applies to reinforcement learning: a pretrained structure generator (acting as the agent) learns to create new structures based on a

reward system provided by a property predictor (serving as the environment), with the objective of maximising the reward.” – Lines 57-67

“One intriguing example is the work of Shen et al., who introduce the concept of “deep dreaming” for the optimisation of organic molecules. This approach begins by training an ML model to predict molecular properties from their string-based representations using only labelled data. Then, through inceptionism — a technique originally developed to visualise the patterns and features learned by convolutional neural networks from images — the ML architecture is inverted to modify the input towards a target property value, effectively creating new molecules in the process. Shen et al.’s findings reveal a few key points: first, [. . .] thereby generating distributions of molecules with systematically shifted properties; second, that inverse design can be achieved in a relatively data-efficient manner by relying solely on labelled datasets to train a regression model, thus eliminating the need for a large-scale pretraining phase; and third, . . .” – Lines 71-84

“To reverse-engineer MOFs, we train separate models to learn a mapping, $f(X) \rightarrow \hat{y}$, between MOF strings (X) and the properties of interest (\hat{y}). For this purpose, we use [. . .] The same architectures are then used to optimise MOF linkers, $f(\hat{y}, y_T) \rightarrow X$, by modifying the edge SBU encodings of a seed structure towards a target property, y_T (Figure 2).” – Lines 194-201

New figures, Figures 1 and 2, have been added to the manuscript as a visual aid for the last modification proposed above (please see **Response 3.4**).

Furthermore, we provide some of the commentary above on our choice of architecture in a revised format in Supplementary Note 3:

“This architectural design allows LSTMs to excel in settings where understanding extended context is crucial, such as in correlating MOF string representations with their material KPIs. We chose this architecture over simpler models, like a fully connected feed forward neural network (as in the work of Shen et al.), as we found they were not capable of correlating MOF strings to their property values with sufficient accuracy. We also ruled out more advanced models like transformers due to the LSTM model’s ability to handle sequences as continuously differentiable inputs.”

And we have added a new Supplementary Note 11 to discuss our choice of representation, which includes Figures R1 and R2 above (identified as Figure S13 and S14 in the new supplementary information), and we refer to this note in the manuscript:

“However, these moieties can be explicitly encoded (and therefore more frequently preserved) by leveraging group tokens for entire chemical substructures, which is an advantage offered by Group SELFIES (see Supplementary Note 1).

While we use SELFIES as the foundation of our MOF string representation, it is important to note that other methods, such as SMILES and DeepSMILES, may also be used to represent molecules in ML applications, as discussed by Krenn et al. The oft-cited advantage of SMILES is its human readability. However, an important limitation is in its tendency to produce strings that do not map to valid molecular graphs, which can be attributed to the fragile grammar. While it is yet to be established whether, in general, SELFIES or SMILES strings yield better ML learning outcomes, we find that SMILES tends to perform better in the forward process of our model. Nevertheless, our focus is on the reverse process, where the syntactic and semantic robustness of SELFIES notation becomes crucial (see Supplementary Note 11).” – Lines 488-501

Comment 1.2 – Adding details on how to perform automated interpretation of the proposed deep dreaming pathways. Since this is the most distinguishing feature of the proposed method, it would be

good for the community to be able to take advantage of that. This can also serve as an example in the code repo.

Response 1.2 – We thank the reviewer for their useful suggestion. In response, we have enhanced our code repository to include new functionality that automates the interpretation and visualisation of deep dreaming pathways. Below, we outline the key additions and improvements:

1. Automated interpretation of the transmutation pathways:
 - The framework now outputs the “valid transmutation pathway”, which consists of chemically valid linker transformations (as defined in the *Optimising Individual MOF Structures* section), and the “full transmutation pathway”, which includes all transformations suggested by the model, valid or not.
 - To aid in interpretation, the framework identifies the **important tokens** that are utilised at intermediate steps to traverse from one valid transmutation to the next. This enables users to uncover design rules employed during optimisation.
2. Visualising the transmutation pathways:
 - We provide methods to chart and interpret the optimisation pathways visually. This helps users to understand how token manipulations manifest in the valid decoded molecules.
3. Statistics and property analysis:
 - We have introduced functionality to compute statistics on token manipulations (e.g., most / least common substitutions, additions, removals). These are accompanied by methods that relate to the properties of the linker (in terms of their molecular weight, solubility, number of hydrogen donors / acceptors, and polarizability).
4. Code repository update:
 - The updated framework includes this suite of new functions under the module “analyse_opt_pathway.py”, stored here. Users can generate detailed statistics, visualise pathways, and explore token-level contributions with minimal setup.

We hope these additions will benefit the community by aiding in the interpretation of the deep dreaming optimisation pathways for various optimisation objectives. We have made the following changes to the manuscript:

*“This pattern aligns with observations from the seed distribution, where MOF structures falling within the medium c_p range feature linkers of moderate molecular weight. **Just like with the VF, these optimisation strategies are also evident in the individual transmutation pathways, examples of which, along with their automated interpretations, are detailed in Supplementary Note 7.**” – Lines 320-325*

And we have provided an example case study in a new Supplementary Note 7:

“To help uncover important design rules, we have provided functionality that automates the interpretation and visualisation of deep dreaming pathways. This is demonstrated in a case study focussed on minimising the c_p of a MOF.

In Figure S7, we show how, in charting a path towards a MOF with lower c_p , the model opts to retain the pyridine substructure of the seed linker while appending bromine atoms to the molecular backbone. This is noted as a consistent strategy in this particular optimisation, where the model identifies the important tokens utilised at steps between valid linker transmutations. The intermediate steps are the collection of modifications suggested by the model that are invalid, according to our definition of linker validity in Supplementary Note 5. In particular, [Br] tokens and nitrogen-containing Group tokens (such as [pyrimidine] and [pyridine]) are considered important in these intermediate pathways.

Importantly, different pathways may emerge from the same seed structure under different noise initialisations. As shown in Figure S8, this pathway favours sulphur and

chlorine additions, while another pathway (in Figure S9) prefers the addition of sulphur followed by bromine. These visualisations illustrate how the same optimisation objective can be achieved through multiple distinct routes.”

We have provided Figures S7 – S9 below.

Figure S8. This figure illustrates an example whereby, in order to minimise the c_p of a seed MOF, the model charts a pathway that considers sulphur and chlorine tokens as important tokens in the first few transmutations, followed by the pyrimidine group token, chlorine, and sulphur SELFIES tokens in the last few transmutations. The importance of these tokens manifest in the decoded structures, whereby we see sulphur-, chlorine-, and nitrogen-containing groups as a common strategy in the optimisation pathway.

Figure S9. This figure illustrates an example whereby, in order to minimise the c_p of a seed MOF, the model charts a pathway that considers sulphur- and nitrogen-containing tokens as important transmutations in the first instance, followed by the importance of bromine tokens in the last transmutation. As in the previous case studies, the importance of these tokens manifest in the valid decoded molecules.

Comment 1.3 – Additional discussion of representations for molecular design [2], as well as the advantages and disadvantages of using text-based representations since the paper mentions that not including geometry leads to modeling shortcomings of certain properties. I would also encourage the authors to look at additional work that has exposed shortcomings of text-based representation in materials design [3] that can influence future work.

References

[1] Ghugare, Raj, et al. "Searching for High-Value Molecules Using Reinforcement Learning and Transformers." The Twelfth International Conference on Learning Representations.

[2] Krenn, Mario, et al. "SELFIES and the future of molecular string representations." Patterns 3.10 (2022).

[3] Alampara, Nawaf, Santiago Miret, and Kevin Maik Jablonka. "MatText: Do Language Models Need More than Text & Scale for Materials Modeling?." AI for Accelerated Materials Design-Vienna 2024.

Response 1.3 – We agree that some additional commentary on our representation, and how our method differs from other approaches (such as structure-based ML models) is warranted, given our discussions in **Response 1.1** and **2.4**. In addition to the modifications outlined in **Response 1.1**, we have made the following modifications to the manuscript:

*“Collectively, these elements capture the chemical and some topological aspects of MOFs, as detailed in the Methods section. **Note that we do not include any structural details here, like the 3D atomic coordinates or geometrical properties of the MOF. While this approach may be considered less expressive than structure-based models such as graph neural networks, it allows for an efficient exploration of the property space through straightforward string manipulations in the reverse process. This strategy therefore avoids the significant computational demands associated with generating and analysing hundreds of MOF structures during deep dreaming.**”* – Lines 167-174

“Given that adsorption properties are highly dependent on the spatial configuration of the atoms — evidenced by the Q_{CO_2} and S_{CO_2/N_2} distributions of the relaxed crystal structures in Supplementary Note 6 — it is not surprising that this choice of representation is less effective at optimising these properties. Recent work by Alampara et al. suggests that including this information may not necessarily enhance performance, as text-based approaches struggle to utilise this data effectively. Instead, they tend to emphasise local chemical environments while neglecting global structural features, leading to challenges in capturing long-range interactions or periodicity. This reflects an inherent limitation of language models in application to material’s design. Nonetheless, the property shifts we observe here are still noteworthy . . .” – Lines 372-380

Reviewer #1 (Remarks on code availability):

Comment 1.4 – The repo is generally well-documented outlining relevant requirements and scripts. I would recommend providing the full dataset for MOF construction to replicate the experiments in the paper, as well as creating or pointing notebooks for the interpretability analysis.

Response 1.4 – We thank the reviewer for their recommendation. We have added all of the CIF files for each case study, along with the dreaming results (stored in .csv file format), for every property explored in our paper in the relevant sub-directories at (https://github.com/SarkisovTeam/dreaming4MOFs/tree/main/deep_dreaming_experiments). There are jupyter notebooks in each subdirectory to reproduce the results in our paper. We have also

included code snippets in the relevant jupyter notebooks to produce the figures in **Response 1.2**. All of the data (including the CIFs of the MOFs used to train our models) and scripts required for training our models are also stored at https://github.com/SarkisovTeam/dreaming4MOFs/tree/main/train_models.

Reviewer #2 (Remarks to the Author):

Remark – This manuscript discusses MOF generation strategy which integrates material property evaluation during the generation such that generated MOFs are only shifted towards certain properties of interest, instead of blindly generating large number of structures. This strategy is largely motivated by the potentially unlimited number of MOFs that can be generated in silico. Authors dub this strategy as "deep dreaming" but this strategy is simpler just inverse design of materials. Having said this, I think the manuscript is an interesting paper that could find potential utility in the ongoing effort in the field to systematically generate hypothetical MOFs, and I support the publication of the manuscript.

Response – We thank the reviewer for their positive feedback on the manuscript. We also appreciate the comments and questions posed in their review; they helped us improve the quality of the work and helped us identify interesting future avenues of research.

Indeed, our focus has been on inverse MOF design. However, the term (and concept) of "deep dreaming" predates our work and was introduced back in 2015 by Google. We hope to clarify (without intending to repeat the entire argument of Shen et al.) with a proposed modification to the manuscript, why we use the term "deep dreaming" specifically:

"Deep dreaming is an experiment that aims to understand how neural networks learn from data. For example, one can train a model to classify images by showing it millions of labelled examples and optimising the network's parameters to achieve accurate classifications. For a task such as this, the network encodes increasingly complex features across layers, from simple textures to shapes of objects, which allows the model to differentiate one image class from another. Once trained, one can visualise what the neural network has learned in these internal layers by fixing the weights and biases of the model, feeding it an image, and then asking it to enhance certain features. This occurs by the process of inceptionism, whereby the network's parameters are no longer updated to minimise the classification error via gradient descent, but rather the input image is modified by maximising the activations of a particular layer via gradient ascent. As each layer is responsible for different levels of abstraction, this process can produce "dream-like" interpretations of the original image, aptly leading to the term "deep dreaming" . . ." – Lines 112-124

Indeed, we can visualise what a model has learned in image classification tasks using deep dreaming, as shown in Figure R4. One can also interpret this process as a generative modelling task, where we are creating a new, enhanced image from a given input. Thus, the idea of training an ML model for some regression / classification tasks, and then utilising the optimised weights and biases of the model in the reverse process for generative tasks, has been coined as "deep dreaming". This terminology aligns with precedents and terms established by previous works, which have inspired our approach. To maintain consistency with these foundational studies and to clearly differentiate our method from other inverse design strategies, we have chosen to retain this terminology in our work.

(a) Training

(b) Deep dreaming

Figure R4. (a) Training a machine learning model for image classification. A typical neural network architecture for this task is composed of: (i) several convolutional layers, which are responsible for the hierarchical feature extraction process; (ii) a fully connected layer, which combines the learned features for the purpose of classification; and (iii) an output softmax activation function, which ensures the classification probabilities properly sum to 1. During training, the network's parameters are updated to minimise the classification losses. (b) Visualising what the neural network has learned after training using inceptionism or “deep dreaming”. In the example provided, the original image is processed by the model, this time with fixed model parameters, and certain patterns are enhanced by maximising the activations of a particular layer. Maximising the activations of shallow convolutional layers (i.e., $\max(a_{L1})$) produce dream-like images with simpler features, while for deeper layers (i.e., $\max(a_{L5})$ or $\max(a_{L9})$) the generated images show more complex enhancements. These images were produced using the code provided at the following webpage.

Comment 2.1 – Is the developed ML model applicable only to MOFs with a single organic linker, or can it also be applied to MOFs containing two or three different linkers?

Response 2.1 – The current implementation is restricted to a single linker type. This is partly motivated by the fact that experimental MOFs are easier to synthesise and control when only one linker type is present, and this is reflected by the prevalent use of single linkers per MOF in the CoRE 2019 MOF database (Figure R5).

Figure R5. Distribution of the number of unique linkers per MOF in the CoRE 2019 MOF database.

However, this approach also simplifies the architectural design of our ML model, which we found to be a suitable starting point for inverse MOF design via the deep dreaming approach. In future iterations, we expect it would be possible to incorporate additional linkers. For example, we could utilise an architecture that handles multiple linkers by representing them as a concatenated one-hot encoded feature vector. Optimisation of both linkers simultaneously could occur by a similar mechanism whereby the full feature vector is updated using gradient information, or even restrict the focus to optimising one linker while keeping the other fixed by “zero-ing” the gradients for the portion of the feature vector representing one of the linkers.

We have made the following modifications to the manuscript to make clear this point:

*“It is therefore important to capture these dependencies simultaneously within our approach. **Additionally, it is worth reiterating that our primary goal is to optimise the edge SBU only, and in this work, we will focus exclusively on MOFs that feature a single unique edge SBU. To achieve this, the edge SBU encoding must be differentiable, enabling our model to modify it using target error gradient information.**”* – Lines 158-161

*“We further discarded any MOFs with cell lengths greater than 60 Å or whose number of atoms exceed 1,500 to minimise the computational cost associated with material property predictions [. . .] **While our ML model is currently tailored to optimise MOFs featuring a single unique edge SBU, we anticipate that future modifications to the architectural design will accommodate MOFs with multiple unique edge SBUs.**”* – Lines 512-518

Comment 2.2 – When optimizing the ligands, does the authors considered only ligands with two connection points? If not, it would be better to include examples of optimizing ligands with three or more connection points.

Response 2.2 – In this study, we focused exclusively on ligands with two connection points, which are classified as edge structural building units (SBUs). This classification aligns with the constructs used in *pormake* as well as the conventions for defining MOF strings within our research framework.

The optimisation of N-connected ligands ($N > 2$), which are classified as node SBUs, was not considered in our work. Introducing these components significantly increases the complexity of the synthetic design, as the additional connection points often introduce asymmetry and can lead to highly unstable structures under operational conditions, unless meticulously designed. Although our current framework does not include ligands with three or more connection points, it presents a valuable avenue for future research. This would involve developing a more advanced model that can balance the chemical functionality and structural stability required for multi-connected ligands. For now, however, we believe this is out-with the scope of the current model framework, which has been tailored to enhance the performance of two-connected edge SBUs.

We have made the following modifications to the manuscript in light of the reviewer’s question:

*“**We extend the methodology described above for the inverse design of MOFs. Our aim is to recover an optimised representation that corresponds to a structure with enhanced properties. However, this task is more challenging as we need to consider not only the edge SBU (the organic linker with two connection points that bridges between node SBUs), but also the node SBU (the inorganic component with more than two connection points), their interactions, and how they coordinate in a given topology.**”* – Lines 150-155

Comment 2.3 – In optimizing MOF properties, where the ligand is optimized, does the model include structural optimization of the MOF? If so, what method was used for the optimization—DFT or MLP-based optimization? The property prediction results may vary depending on whether the MOF’s structure has been optimized. If it is too cumbersome to include this as part of the manuscript, it would be good to add a brief discussion of potentially integrating this as part of the workflow.

Response 2.3 – We appreciate the reviewer's insight regarding the importance of structural optimisation in the design of MOFs. The results presented in the manuscript are obtained without this component. However, we recognise the critical role of this step for the realistic development of MOFs designed *in silico*. Indeed, any computational design of MOFs would ideally be followed by structural relaxation to ensure that the predicted properties are reflective of a physically stable structure. This is particularly crucial before proceeding to experimental synthesis and testing of interesting candidates. While the objective of the work was not necessarily to find such a candidate, we agree it is necessary to provide some commentary on this important post-processing step, which would form an integral part of the development pipeline for inverse MOF design. We have therefore included in Supplementary Note 6 a subsection that investigates the impact of structural relaxation on the property distributions obtained from our deep dreaming model:

“Given the hypothetical nature of the generated MOFs, their crystalline structures are unlikely to be perfect. Therefore, structural relaxation is generally required to optimise their atomic coordinates and achieve more realistic configurations. Although the results in the main article are derived from unrelaxed structures to initially showcase the deep dreaming approach for inverse MOF design, incorporating this additional post-processing step will likely be required before translating hypothetical MOFs into real materials.

To evaluate the impact of this optimisation on MOF properties, we selected a representative subset of $n = 100$ MOFs from the distributions presented in Figure S5 (of size $n = 1,000$). We fitted a probability density function to these distributions and drew samples that retained the features of the original datasets. These MOFs were then structurally optimised using the UFF forcefield as implemented in the Forcite Module of Materials Studio 2019, with property values recalculated as outlined in the MOF property determination section of the main article.

In the case studies for optimising c_p , our findings reveal that structural relaxation has a minimal impact on the overall results. The mean absolute error (MAE) between the relaxed and unrelaxed structures is approximately $0.03 \text{ J g}^{-1} \text{ K}^{-1}$, indicating a deviation of around 3.75%. This comparison is detailed in Figure S6. However, more significant differences are observed in other parameters, including the bandgap, Q_{CO_2} , $S_{\text{CO}_2/\text{N}_2}$, and gravitational surface area (GSA). The respective MAEs for these parameters are about 0.64 eV, 5.94 kJ mol^{-1} , 0.79, and 288.5 $\text{m}^2 \text{ g}^{-1}$, translating to average deviations ranging from 15% to 30%. These differences are comparable to the prediction errors of our ML models, as reported in Table S7. Most notably, the MAE for pore volume (VF), as shown in Figure S6, is 0.27, which is approximately five times greater than the prediction errors in Table S7. Evidently, structural relaxation plays a critical role in accurately evaluating properties that solely rely on the exact atomic coordinates (i.e., VF), an important role for other properties that depend on both the chemistry and structural features (i.e., the bandgap, Q_{CO_2} , and $S_{\text{CO}_2/\text{N}_2}$), and a minimal role for properties predominantly determined by the material's chemistry, such as the c_p .”

We have provided the newly added Figure S6 below for the reviewer's reference. Furthermore, we refer to this note in the manuscript at different points:

“Taking only the best-predicted structure from an aggregate of 10 local optimisations, we reconstruct MOFs from their encoded representations without relaxing the crystal structures (see the Dataset curation section) and recompute the “ground truth” c_p values using the ML model of Moosavi et al. The results are displayed in Figure 5 (a). [. . .] These results are shown to be insensitive to the exact atomic coordinates of the MOFs, as detailed in Supplementary Note 6, where crystal structure relaxation has minimal impact on the property distributions. This successful outcome is facilitated by the design of linkers that closely reflects the physics of c_p itself . . . ” – Lines 296-305

“Given that adsorption properties are highly dependent on the spatial configuration of the atoms — evidenced by the Q_{CO_2} and S_{CO_2/N_2} distributions of the relaxed crystal structures in Supplementary Note 6 — it is not surprising that this choice of representation is less effective at optimising these properties.” – Lines 372-375

Figure S6. Impact of structural optimisation on MOF property distributions. We strategically sampled 100 MOFs (middle column) from the original dreamed distributions (left column), which are obtained from unrelaxed structures. We show how these sampled distributions are affected by structural relaxation in the right column.

Comment 2.4 – As shown in the property predictions in Table S7, the model shows poor predictions for adsorption-related properties such as heat of adsorption and CO₂/N₂ selectivity. Does this mean that MOF optimization using this model cannot be applied to improving adsorption performance and is only applicable to texture properties? Is there a specific reason why the author did not include 3D coordinate information to improve the prediction of those?

Response 2.4 – We demonstrate that textual representations alone are not as proficient at predicting properties that are highly dependent on the 3D configuration of atoms, as in the case of Q_{CO_2} and S_{CO_2/N_2} . While this reflects an inherent shortcoming of language models compared to structure-based ML models specifically tailored for these tasks – see, for example, the work of Cao et al. (2023) – this approach presents a different advantage over structure-based ML models. Structure-based ML models require the 3D coordinates of the MOF to obtain the relevant features for property prediction, whereas text-based language models do not. This becomes particularly useful in inverse design, as we can explore the property space more efficiently through simple manipulations of the molecular strings. This contrasts with using features based on the 3D coordinates of MOF atoms, which is much more cumbersome, particularly when the number of atoms in the unit cell become large (on the order $10^2 - 10^3$ atoms). To give an example, a single deep dreaming experiment can evaluate anywhere between 500 – 1,500 MOF strings before arriving to the final optimised representation. This process only takes a couple of minutes on a standard CPU (using the default optimisation settings detailed in the supporting information). However, a structure-based approach would require one to construct the CIF file for each of these MOF strings and compute the relevant features before obtaining a property estimate for subsequent iterations of the optimisation. For this reason, we forgo including these features in the forward (regression) process, which allows us to quickly perform inverse design in the reverse (dreaming) process.

Despite the absence of this structural information, our model's predictive performance for Q_{CO_2} remains competitive when compared to other similar models built upon textual representations. For example, the transformer model of Park et al. (2024), which also uses textual representations, reports a mean absolute error (MAE) for Q_{CO_2} of 2.87 kJ/mol using approximately 33,000 MOFs for training/testing/validation. In contrast, our model achieves an MAE of 4.14 kJ/mol with a smaller dataset of 12,000 MOFs for training/testing/validation, indicating that our model is still effective within its operational constraints. This predictive capability manifests in the inverse process, where we observe notable shifts in the property distributions for Q_{CO_2} .

Of course, there is opportunity to further improve the performance of the model by including more training datapoints or by performing hyperparameter optimisation. For example, the learning curve for

Figure R6. Learning curves for the deep dreaming model out-of-sample predictive performance for Q_{CO_2} , in coordinates of mean absolute error (red y-axis), coefficient of determination (blue y-axis), and training sample size (x-axis). The models are trained on subsets of the full training set, while the validation and test sets are kept constant. The training subset sizes are 100, 200, 500, 1,000, 2,000, 5,000, and 10,000.

Q_{CO_2} in Figure R6 indicates that more training data (beyond 10,000 data points) would likely return better results than what is reported in the current manuscript. However, we found that 10,000 training points allowed us to develop a model that is sufficiently predictive for the purpose of exploring deep dreaming as a concept.

In light of the reviewer's question, and the suggestion of reviewer 1 in **Comment 1.3**, we have made the following modification to the manuscript:

*“Collectively, these elements capture the chemical and some topological aspects of MOFs, as detailed in the Methods section. **Note that we do not include any structural details here, like the 3D atomic coordinates or geometrical properties of the MOF. While this approach may be considered less expressive than structure-based models such as graph neural networks, it allows for an efficient exploration of the property space through straightforward string manipulations in the reverse process. This strategy therefore avoids the significant computational demands associated with generating and analysing hundreds of MOF structures during deep dreaming.**”* – Lines 167-174

We have also added in Supplementary Note 4 the following change:

*“For the bandgap, Q_{CO_2} , and S_{CO_2/N_2} , the agreement between “ground truth” data and the model predictions are more modest. **Learning curves for these properties indicate that adding more training data would likely lead to better performance. However, we found 10,000 training points to be acceptable for the purposes of our investigation. Parity plots of the model predictions are provided in Figure S3.**”*

Comment 2.5 – It would be better for the author to include prediction results for other texture properties such as surface area, pore size, etc.

Response 2.5 – We thank the reviewer for their suggestion. While we do not show all the results in the manuscript (in the interest of brevity) we have included properties that depend on the structure (i.e., the surface area and the pore volume), chemistry (specific heat capacity), or some combination thereof (heat of CO₂ adsorption at infinite dilution, CO₂ / N₂ Henry selectivity, and bandgap), in our complete analysis. We believe the results on the pore volume and surface area relate to what the reviewer is referring to, i.e., the textural properties. The regression results for all these properties may be found in Table S7 of the revised supporting information, and the deep dreaming results may be found in Figure S5. Our belief is that these properties are sufficiently diverse to probe different facets of the model's capabilities in the context of inverse MOF design for different applications, and we refer the reader to these results for their reference in the section titled *Shifting property distributions* of the revised manuscript:

“Thus, examining the cp of MOFs in relation to carbon capture processes provides an excellent benchmark to probe the capabilities of the model in a setting where one might be interested in either minimising or maximising the property value, depending on the target application (however, for completeness, the dreaming results for other MOF properties are provided in Supplementary Note 6).” – Lines 288-292

Comment 2.6 – In the MOF optimization, does the model also consider ligand functionalization, such as functional group substitution? This is known to have a significant impact on improving adsorption performance.

Response 2.6 – The reviewer raises a very interesting question. Our model operates on string-based representations of MOFs that include descriptions of both the backbone and the functional groups of the organic linkers. This allows the model to explore a variety of functionalities by modifying these strings. For example, we can simulate functional group substitutions by replacing specific tokens in a MOF string (e.g., replacing [Cl] with [F], [O], etc.) as demonstrated in Figure R7. This figure shows how the specific heat capacity values change with respect to these modifications.

Figure R7. MOF linker functional group substitution. In this experiment, we substitute the [Cl] token in the edge SBU string of a seed MOF (left) with other functional group tokens. Our ML model, trained to predict the c_p of MOFs from their textual representations, predicts how the properties of the MOF change with these substitutions (right). The values above each structure show the ground-truth data in closed brackets, derived from constructing and relaxing the corresponding CIF files, alongside the ML model's predictions outside the brackets.

However, this experiment – effectively a simple string manipulation – does not fall under “deep dreaming” as we have defined it in our work. Strictly speaking, we refer to “deep dreaming” as the continuous optimisation of the entire encoded representation of the edge SBU using target error gradient information. This means that functional group substitution occurs in our approach naturally if this is where the gradient is pointing, but we do not orchestrate it manually (while retaining the topological features of the seed linker).

However, the reviewer’s question points towards a potential refinement in our approach. Currently, our deep dreaming model could be extended to fix portions of the edge SBU encoding to focus changes on functional groups only. This concept is touched upon in our discussions on multi-linker optimisation in **Response 2.4**, although this would require adjustments to our framework to control for token interpretation. To clarify, in Figure R7, we sampled only a subset of tokens representing the functional groups of interest. However, in a comprehensive deep dreaming experiment, the entire edge SBU vocabulary could be sampled during a focussed optimisation. We therefore repeated the experiment in Figure R7, instead substituting the [Cl] token with random tokens from the complete edge SBU vocabulary. This experiment acts a proxy demonstration of the deep dreaming process, whereby the seed structure may be subject to multiple substitutions during optimisation. The results are shown in Figure R8, whereby we recover some structures that achieve a functional group substitution as desired, while others significantly diverge from the scaffold of the seed. This divergence may be attributed to the context-dependent nature of (Group) SELFIES tokens, which can represent different chemical entities or structural features depending on their position in the string. While this particular

Figure R8. Proxy demonstration of a “focussed” deep dreaming experiment. The task here is to modify the chlorine functional group of the seed structure, while keeping the rest of the molecule fixed. For a seed MOF linker (left), we randomly substitute the [Cl] token – the functional group we wish to modify – with a token from the full edge SBU alphabet over 10 independent runs. On the right, we show how these token substitutions affect the molecular structure of the seed linker.

feature of (Group) SELFIES is what imbues it with 100% chemical robustness¹, it also makes it challenging to maintain specific molecular scaffolds during substitutions.

Going forward, we are interested in refining our method to allow for more precise control over functional group substitutions while retaining the topological features of the original scaffold. This refinement aligns with the research objectives of the Group SELFIES developers, who have suggested that future versions of the Group SELFIES representation may include methods to optimise atom types while retaining specific molecular scaffolds. We expect that developments in this direction will facilitate a more targeted approach to functional group substitution, which we hope will enhance our model’s utility for specific applications like improving adsorption performance.

While we are unable to include all of these discussions in the manuscript in the interest of brevity, we allude to these potential developments in the closing statement of our article:

*“Our results show that, indeed, a model’s learned representations can be leveraged to reverse-engineer new MOF structures, consolidating the tasks of structure optimisation and property prediction within a single unified framework. **We anticipate that future developments in the Group SELFIES algorithm³⁹ and refinements in our methodology will enable more targeted optimisation strategies, such as functional group substitution, providing deeper insights into the MOF structure-property relationship at an early stage.**” – Lines 471-474*

Reviewer #3 (Remarks to the Author):

Remark – In this manuscript, Cleeton, et al. utilizes inceptionism for inverse design of MOF linkers. They train a LSTM enhanced with attention between node and edge embeddings, together with MLP, on synthetic MOF data for this purpose. The paper, for most part, was written clearly, though there are

¹ To satisfy syntactic robustness, (Group) SELFIES tokens are *overloaded*, meaning a token can be used to define a chemical entity or can be used to define the length of a branch, for example. The token ‘[O]’ may be interpreted as an oxygen atom or it may be interpreted as a branch of length 10, depending on whether it appears directly after a branch or ring token in a SELFIES string. For Group tokens, a similar overloading takes place when interpreting the attachment points of the group: an ‘[O]’ token after a Group token is interpreted as a *relative index* of 6, meaning that subsequent tokens after ‘[O]’ would be generated from the 6th attachment point index relative to the initial Group token attachment point. SELFIES also implements a series of derivation rules to satisfy semantic constraints (i.e., valency) and generate valid molecules. In deriving SMILES molecules from (Group) SELFIES strings, the (Group) SELFIES string is traversed, and each token is interpreted as a rule vector that depends on the state of the derivation. For example, a ‘[=O]’ token may be translated to a ‘=O’ or ‘O’ SMILES token (i.e., an oxygen atom which shares two pairs of electrons or one pair of electrons with another atom), depending on the preceding tokens.

several missing details (see comments). While the problem is important, I did not find the work comprehensive in terms of comparing its performance over existing baselines or in terms of validation of optimized molecules. Neither I found the technology leveraged is novel enough. Therefore, my recommendation is that the paper, as it is, is not suitable for publication in *Nature Communications*. I have listed my comments below, which I believe can be helpful for improving the work.

Response – The authors would like to thank the reviewer for their assessment of the work presented. We understand that manuscripts submitted to *Nature Communications* must be sufficiently rigorous, impactful, and interesting to align with the journal's status and broad readership. We also welcome any and all constructive criticism to help improve the quality of our work.

We appreciate the opportunity to clarify and expand on the unique contributions of our work and address the concerns raised. Here (and in the responses to the reviewer's individual comments), we would like to provide some additional arguments as to why we believe our study is suitable for publication:

1. Novelty

- Successfully deploying generative models for the inverse design of materials, and in particular MOFs, is itself a very novel concept. While the idea of generating MOFs with desired functionalities through inverse design has long been sought after, there are only a handful of studies in this domain. Our work contributes to this limited portfolio, offering a new approach for gradient-based optimisation within the real chemical space – something that, to the best of our knowledge, has not been explored before. We believe that new information towards this objective helps the community move closer towards the development of tools refined for creating MOFs with specific, desired properties. Indeed, two other reviewers have found our work very novel, and we hope that our revised manuscript aligns with the reviewer's expectations.

2. Practical utility of the technology

- Our use of LSTMs, while not state-of-the-art in some ML fields, is practically motivated and well-suited for this application. Unlike transformers, LSTMs can operate on differentiable floating-point vectors, enabling the use of gradient information to update input representations. This allows for efficient optimisation of MOF linkers in a manner that is less cumbersome to train and aligns with the data requirements of our study. Following the ML rule-of-thumb of using the simplest effective model, we have chosen LSTMs for their appropriateness to our specific objectives. At the interest of the reviewer, please see **Response 1.1** for some additional discussions on our motivation for the deep dreaming approach.
- Furthermore, the context length for MOF strings are well within the capabilities of LSTMs, which remain highly relevant and competitive in domains like time series forecasting.

3. Comparison

- We appreciate the reviewer for their concerns regarding benchmarking our approach against other generative modelling methods and its practical importance. The point the reviewer raises is important for the whole MOF community. Indeed, recently we have seen new approaches emerge towards inverse engineering hypothetical MOF structures. We discuss a few of these approaches in our work.
- While such a diversity of these models empowers explorative MOF discovery, it comes with a cost: it is not clear at this stage of research how to systematically compare these models to each other and what kind of properties should be used as the benchmark. They differ significantly in the size of the data set they use, in the strategies on how the material landscape is explored, and in the properties of the produced materials. The two approaches (e.g., a VAE and a more local model, developed by us) will arrive to two different sets of materials and, again, given that both sets have achieved the optimisation target, beyond this there is no concrete criteria to compare these sets to each other as this will depend on the chosen metrics

such as synthesisability, diversity, and on how one defines 'validity' (which, in itself, is not a clear definition within the MOF community).

- Not surprisingly, there is no study (to the best of our knowledge) that attempts to provide such a cross model, comprehensive comparison. It seems to us that this field is at the stage of proposing conceptual ideas, with different capabilities, remits, and efficiencies. Unlike other fields of science (such as small organic molecules, which can rely on standardised datasets and frameworks like MOSES, or inorganic crystals who can sample from well-defined databases such as the Materials Project dataset of stable crystals) where generative approaches matured enough to have standardised approaches to performance evaluation, the MOF field is simply not there yet, and this will require some thinking.
- Evolution towards a consistent, general framework to compare the models for inverse MOF design is a significant undertaking, and we think it would require the contribution of several key groups in the field to agree on how to do it consistently.
- In our contribution, we offered a new concept on how an ML model that has been already trained for property predictions can be used to generate new materials with targeted distributions in a relatively data-efficient manner. The practical utility of our approach is significant: many groups develop ML models to predict properties of materials; we open an opportunity for all these groups to generate more materials with targeted characteristics within the same model by simply reversing it, provided they featurise their materials as molecular strings.

We believe it will take some time for the community to figure out unique application niches and usability for each of the emerging methods – and it may turn out that they are not in competition with each other but rather complementary. We expect that our benchmarking efforts and qualitative comparisons in Supplementary Note 10 will help in this regard.

Comment 3.1 – It is not very clear what is the advantage of the proposed method. Is it the ability to train on small number of samples? Is that a big constraint when the data is synthetic?

Response 3.1 – The primary advantage of the proposed method lies in its interpretability. This comes from two perspectives. First, by observing the sequence of molecular modifications executed by the model to achieve its design objective, researchers can derive actionable design rules to enhance specific properties. This is facilitated by the gradient-based optimisation approach, which operates directly within the real chemical space. As discussed in our **Response 1.2**, we provide additional functionality to automatically interpret the transmutation pathways executed by the model, allowing one to visualise intermediate transmutation steps and identify key structural modifications. With growing interest in explainable AI (Rudin (2019), Wellawatte et al. (2023)) we believe our method represents an important contribution to this domain. Second, incorporating attention mechanisms into the model enables the interpretation of token importance during optimisation. When combined with the Group SELFIES representation of linkers, this allows for the identification of atom types or full chemical substructures that are critical for a specific design task. For instance, in the Q_{CO_2} case study, the model consistently identifies aromatic groups with electrophilic character as more important than non-polar groups, providing valuable insights into structure-property relationships.

However, the training requirements are also a benefit. This comes (mostly) from the fact that we avoid the computationally intensive large-scale pre-training typically required by other generative models. Even in the case of synthetic training data, this stage can be laborious. To illustrate this point, we train our models using 12,000 MOFs, primarily composed of atoms in the range of 0 – 500 atoms per unit cell. Structural relaxation for MOFs of this size takes approximately 10 seconds per structure, on

Figure R9. A benchmark on the computational time required to structurally relax MOF structures as a function of the number of atoms in their unit cell. Here, we optimise the crystal structures of MOFs using the UFF forcefield as implemented in the Forcite Module of Materials Studio 2019. The histogram shows the frequency of the number of atoms contained within the unit cells of the 12,000 MOFs used to train/validate/test our deep dreaming models. The red line-plot shows the time taken to relax MOF structures containing 50, 100, 200, 500, 1,000, 2,000, and 5,039 atoms in their unit cells.

average (Figure R9). By contrast, the large-scale pre-training stage of other generative modelling approaches usually require millions of unlabelled data points. Based on similar computational trends, generating unlabelled training data for 2 million structures – approximately the number of MOFs used to train the RL + transformer model of Park et al., for example – would require around 230 additional days of compute time.

Our proposed modifications to the manuscript in response to the reviewer’s comments here are the same as our proposed modifications in **Response 1.1**, and we direct the reviewer to the changes outlined there.

Comment 3.2 – Can the authors use the code in the available GitHub repo to train a Sm-VAE that uses the synthetic data used in this study for the semi-supervised part of the training. That way, it will be possible provide a vis-a-vis comparison on property-optimized molecules using the proposed method and Sm-VAE.

Response 3.2 – We thank the reviewer for their enquiry here. Comparing our proposed method with Sm-VAE using the synthetic data from this study would indeed be valuable. This was one of our initial objectives when we sought to compare our approach with precedent models in revised Supplementary Note 10 of the revised manuscript. However, we encountered significant challenges that make such a direct comparison infeasible with the current resources and code provided in the GitHub repository. Below, we enumerate the primary challenges, not as a criticism of the excellent work by Yao et al., but to highlight the difficulties of implementing such a comparison:

1. Lack of encoding and decoding functionality for MOFs
 - To the best of our knowledge, the Sm-VAE GitHub repository does not include functionality to encode CIF files into RF codes (the string representation Sm-VAE is trained on) or to reconstruct CIF files from RF codes. When Yao et al. provide code snippets on “reconstructing” MOFs within the GitHub repo, it refers to the RF codes themselves, and not the actual CIF files. Although their manuscript presents results involving 3D coordinate information, explicit details of this reconstruction step are absent from both their manuscript and the repository. Without this functionality, performing a direct comparison is not possible.
2. Unclear node SBU naming conventions
 - The Sm-VAE training data includes RF codes that use categorical strings to represent node SBUs, with naming conventions like “sym_7_mc_4” and “sym_3_vae_289.”

However, the corresponding CIF files or explicit mappings between these categorical strings and their associated building blocks are not provided. While visual representations of the building blocks are available in their supporting information, they are not identified with any labels that are compatible with the constructs of Sm-VAE. The absence of CIF files or mappings makes it impossible for us to manually extract or recreate the necessary building blocks for training or testing.

3. Incompatibility between training data

- A fundamental challenge lies in the differences between the training data used in our method and that of Sm-VAE. Sm-VAE utilises a limited set of 14 unique metal nodes in its training data, whereas our dataset includes over 200 unique metal nodes. This discrepancy means that many of the MOFs in our dataset fall outside the applicability domain of the Sm-VAE model. While it is theoretically possible to retrain our models using only the Sm-VAE-compatible metal nodes, we would still face the aforementioned issues of identifying and mapping building blocks or reconstructing MOFs from RF codes.

We recognise the importance of benchmarking in developing new methods for inverse MOF design and agree that such comparisons are crucial for advancing the field. However, achieving a meaningful and accurate comparison would require addressing the current gaps in tools, training data, and methodology. As we mentioned in our opening response to the reviewer, we believe such an endeavour is best approached as a collaborative effort involving multiple research groups to ensure fairness and broad applicability. This is part of what motivated our opening statements in the *Discussion* section of our manuscript, specifically:

“Currently, there is no consistent framework or benchmark data to facilitate this comparison. In fact, given the heterogeneity of the models (the methods and the data they use), comparing their performance on a consistent basis is very difficult, representing a fundamental challenge for the entire field of ML for MOF discovery.” – Lines 401-404

An ulterior objective of our work would be to encourage the community to join forces in building a robust benchmarking framework for MOF generative models, and we look forward to opportunities for collaboration that will advance the development of tools for inverse MOF design.

Comment 3.3 – Can the authors provide an equation that describes the full training loss? From the description, there seems to be a property prediction loss plus a penalty on invalid linkers; however, no further details have been provided.

Response 3.3 – We thank the reviewer for pointing out this oversight. We have included a description of this penalty term in Supplementary Note 5 (where we discuss linker validity constraints), for the sake of completeness. The modifications to the SI are as follows:

*“By design, all molecules generated during deep dreaming are chemically valid, however they may not necessarily be valid from a linker chemistry point of view. Two connection points are required to form bonds with the node SBUs in a given topology. **To facilitate the formation of these connection points, we include an additional loss term to the target losses (computed as the mean squared target error) during the deep dreaming process. As each token position is represented by a probability distribution across the edge SBU vocabulary, we can represent the “probability mass” of each token as the sum of their continuous, noisy one-hot activations in each token position, which we will denote p_{token} . We focus on the token representing a connection site – labelled as [FrH0] in our vocabulary – such that the p_{Fr} allocated to this token across the sequence corresponds to the effective number of connection points. Rather than enforcing a hard constraint by decoding to a discrete molecule and counting the exact number of francium pseudoatoms, we introduce a smooth penalty based on a “soft count” of [FrH0]. Specifically, we apply a quadratic penalty, $\alpha(p_{Fr} - 2)^2$, to discourage deviations from the desired count of two connection points. Because p_{Fr} remains a real-valued,***

differentiable function of the underlying continuous representation, the gradient is well-defined even when p_{Fr} is near 2. This preserves a smooth gradient signal throughout deep dreaming, while still yielding valid, integer-valued molecular compositions upon decoding.

It is also important that these connection points do not arise too close together to avoid the formation of unphysical structures. . .”

Comment 3.4 – The authors should consider a figure to illustrate the forward learning and inverse training, as in the original Shen, et al. paper to help reader understand the generative model. Current writing leaves it ambiguous.

Response 3.4 – We thank the reviewer for their suggestion. We have updated Figure 1 (provided below) to show how the edge SBU is encoded using noisy one-hot vectors. We have also included an additional Figure 2 in the revised manuscript (depicted below, also) that aims to provide greater clarity on how the deep dreaming process occurs via our model, whereby the edge SBU vector representation is updated using gradient information after training, while keeping the weights and biases of the model fixed.

We have also made modifications to the manuscript in several places, which we hope improves the readability of the manuscript:

“Deep dreaming is an experiment that aims to understand how neural networks learn from data. For example, one can train a model to classify images by showing it millions of labelled examples and optimising the network’s parameters to achieve accurate classifications. For a task such as this, the network encodes increasingly complex features across layers, from simple textures to shapes of objects, which allows the model to differentiate one image class from another. Once trained, one can visualise what the neural network has learned in these internal layers by fixing the weights and biases of the model, feeding it an image, and then asking it to enhance certain features. This occurs by the process of inceptionism, whereby the network’s parameters are no longer updated to minimise the classification error via gradient descent, but rather the input image is modified by maximising the activations of a particular layer via gradient ascent. As each layer is responsible for different levels of abstraction, this process can produce “dream-like” interpretations of the original image, aptly leading to the term “deep dreaming”.

Shen et al. leveraged this idea for the inverse design of small organic molecules. However, instead of learning to classify images, they trained a neural network model to predict real-valued properties of molecules from their SELFIES string representations. To learn from these non-numerical features effectively, the SELFIES strings were one-hot encoded — a process that converts each SELFIES token into a binary vector. This vector has a ‘1’ in the position corresponding to the token and ‘0’ elsewhere, ensuring that each character is uniquely represented. Then, by adding noise to the zero-elements, they transformed the input from a collection of binary vectors into differentiable probability distributions over the SELFIES tokens. The rationale for this final step becomes clear in the discussion below.

During training — the forward process — the inputs (noisy one-hot vectors) and outputs (property values) remain fixed while the weights and biases of the ML model are incrementally updated via backpropagation to minimise the prediction errors. In molecular deep dreaming, or the reverse process, this paradigm is inverted: the pretrained weights and biases of the model are frozen, and the input is incrementally modified towards a new, optimal feature vector using gradient descent. This is achieved by first computing the target error, which is the error between the predicted property of a molecule and some target property. Then, the gradient of the target error with respect to the encoded input is

calculated, and the initial molecular structure is modified via backpropagation such that the target error is minimised. **Gradient-based molecular optimisation such as this is only possible because the molecule is represented by a differentiable probability distribution over SELFIES tokens. This highlights the importance of noisy one-hot encoding in facilitating deep dreaming experiments.** As SELFIES is 100% chemically robust, the optimised representation will always decode to a valid molecule and, provided the mapping between molecular strings and their properties is sufficiently accurate, will return a structure with improved functionality.” – Lines 112-148

“The model integrates Long-Short Term Memory (LSTM) networks, augmented with attention mechanisms, alongside conventional multilayer perceptron (MLP) neural networks. **To prepare the MOF strings as input, they are first decomposed into differentiable (edge) and non-differentiable (node and topology) components. The edge is encoded using a noisy one-hot vector, similar to the work of Shen et al. (Figure 1 (b)). Since the node and topology representations remain fixed during deep dreaming, we utilise more advanced token embeddings to encode their contributions. We then process the differentiable and non-differentiable portions of the MOF strings using two distinct LSTM branches.**” – Lines 176-184

“Figure 1. Deep dreaming model and MOF string representation. (a) Deep dreaming modelling approach. MOFs are decomposed into an edge SBU, node SBU, and a topology, which we characterise using a string-based representation. A bespoke sequence-to-regression machine learning model, composed of LSTM cells, attention mechanisms, and multilayer perceptrons, are then used to map MOF strings to their respective material properties. To facilitate the optimisation of MOF linkers via reverse differentiation of the input, the linkers are encoded using a one-hot encoding with random noise added to the zero-elements. To capture contributions from the node and topology, we use token embeddings. Connection points between SBUs are identified using francium pseudoatoms in the MOF string representation. (b) Encoding MOF strings. Edge SBUs are represented as SELFIES strings, which can be converted into a machine-readable feature vector by using a one-hot encoding, which represents tokens as binary vectors. Random noise is then added to the zero-elements of the one-hot encoded representations, effectively transforming them from a simple bit vector into a differentiable probability distribution over SELFIES tokens. The complete MOF string encoding is then obtained by concatenating the edge, node, and topology encodings.”

“Figure 2. Reverse engineering MOFs using deep dreaming. (Left) In the forward (training) phase, the machine learning model learns to predict the properties of MOFs from their (fixed) encoded representations by adjusting the model weights and biases through a standard feedforward and backpropagation process to minimise the prediction errors. (Right) In the reverse (dreaming) phase, the same feedforward and backpropagation process is used to update the edge SBU encoding – keeping the weights and biases of the model fixed – to minimise the difference between the predicted property of an initial MOF structure and some target property. The sequence of modifications executed by the model, from the initial MOF structure to the final optimised MOF structure, forms the optimisation pathway.”

Comment 3.5 – As the results suggest that generated molecules are highly novel (low SNN) with respect to training distribution, how do we know that the predictive model generalize reliably on this novel distribution?

Response 3.5 – We thank the reviewer for raising an important concern regarding the generalisability of our predictive model. Indeed, our approach utilises a regression model to reverse engineer new MOFs, and we recognise that samples generated from regions distant from the training manifold might yield less accurate predictions. This extrapolation challenge is common in machine learning and warrants cautious handling.

The results presented in the manuscript are the “ground-truth” values, i.e., the property values have been computed using high-fidelity simulation or machine learning surrogates trained on quantum-mechanical data. As such, what we show is that deep dreaming can achieve notable shifts in the property distributions far from the central tendency of the seed distributions, based on rigorous simulation approaches. However, we have provided some guidance on how to handle these concerns in a new Supplementary Note 8:

“Deep dreaming relies on using a regression model to reverse engineer new MOFs. These architectures are typically designed to interpolate within the range of data they have been exposed to during training. When tasked with predicting properties or generating structures that are outside this range, there will invariably be some challenges with extrapolation.

To illustrate the novelty of the linkers optimised for the c_p case study in the main article (Shifting property distributions section), we employed a t-SNE projection to visualise the chemical space coverage relative to the training set. This revealed that our dreamed linkers can map to points far from the training manifold. Here, we group MOFs in the dreamed distributions based on the Tanimoto similarity of the linker to their nearest neighbour linker in the training set and computed the average prediction errors within each group. Our findings, depicted in Figure S10, indicate that MOFs with lower Tanimoto similarity (i.e., with features more dissimilar to the training set) exhibit greater prediction errors compared to those with higher similarity. This observation highlights the regions where the model’s predictions are less reliable.

To address the issue of generalisation, we can adopt “uncertainty-aware” deep dreaming models by implementing an uncertainty quantification approach. For example, we train 10 deep dreaming models using different training sets, resampled with replacement, and estimate the uncertainty in our model using the variance in the predictions of the ensemble. By grouping these uncertainty estimates by Tanimoto similarities, similar to Figure S10, we find that this method is fairly effective at differentiating between regions of high and low uncertainty (Figure S11). This additional layer of uncertainty quantification provides valuable insights into the model’s reliability across various chemical spaces. It also supports a more informed exploration of novel chemical spaces through an iterative generative and training feedback loop. Specifically, samples identified with high uncertainty could be prioritised for inclusion in the training set in subsequent model iterations, enhancing the model’s generalisability. By incorporating these strategies, practitioners may expand the applicability beyond the initial training domain, which, in turn, can facilitate more reliable MOF design.”

We have provided Figures S10 and S11 below. Furthermore, we direct interested readers to this note in the manuscript with the following modification:

“To corroborate this, we conducted an experiment where we randomly sampled (without replacement) 1,000 linkers from our training set and compared them to the rest. The result, an SNN value of 0.45 ± 0.001 (with the standard deviation calculated over five iterations),

confirms that the generated linkers are more dissimilar from the training manifold than would be observed by a random sampling of existing structures. **This point is further illustrated in Figure 6, where we show the dreamed distributions in 2D phase space relative to the training set, clearly demonstrating that the model is charting largely unexplored areas of the chemical phase space.**

As we utilise pretrained weights and biases for deep dreaming, another question arises on whether the model is capable of generalising to structures far removed from the training set. We do observe greater prediction errors for MOFs composed of linkers that are highly dissimilar to those observed during training. However, by quantifying the uncertainty in our model predictions, we can gauge where reliable generalisations are likely as we chart these unknown areas (see Supplementary Note 8).” – Lines 339-353

Figure S10. Prediction mean absolute error (MAE) for c_p -optimised MOFs, grouped according to the Tanimoto similarity of their dreamed linkers relative to their nearest neighbour linker in the training dataset. MOFs with lower Tanimoto similarities show higher prediction MAEs.

Figure S11. The uncertainty in the deep dreaming model predictions, grouped according to the Tanimoto similarity of their dreamed linkers relative to their nearest neighbour linker in the training dataset.

Authors' Comments

Throughout the following responses, we use double quoted italic text in green ("*example*") to indicate text taken from the originally submitted article and supplementary information, and bolded green text to indicate proposed changes for the revised article and supplementary information ("***modified example***"). Our response to reviewers' comments will be indicated by normal text coloured blue (example), while original comments by the reviewers will be indicated by normal text coloured black (example).

Reviewer #1 (Remarks to the Author):

Remark - I thank the authors for their response and edits to the manuscript. I think the manuscript clarified much of the initial feedback. I am in favor of supporting the manuscript, assuming the following updates can be made:

Comment 1.1 – Adding additional references on applying reinforcement learning type techniques to materials design, including MOF (ref. 8 is already included), molecules [1], proteins [2], and crystal structures [3], to line 67 clarifying the procedure: "A similar principle applies to reinforcement learning: a pretrained structure generator (acting as the agent) learns to create new structures based on a reward system provided by a property predictor (serving as the environment), with the objective of maximizing the reward."

[1] Ghugare, Raj, et al. "Searching for High-Value Molecules Using Reinforcement Learning and Transformers." The Twelfth International Conference on Learning Representations.

[2] Lutz, Isaac D., et al. "Top-down design of protein architectures with reinforcement learning." Science 380.6642 (2023): 266-273.

[3] Govindarajan, Prashant, et al. "Learning conditional policies for crystal design using offline reinforcement learning." Digital Discovery 3.4 (2024): 769-785.

Response 1.1 – We thank the reviewer for their feedback in the revisions of the manuscript. We have included the references suggested above in our manuscript at the appropriate location.

Comment 1.2 – Adding a discussion on the importance and difficulties of doing proper benchmark for ML-based materials design. This could be done in the Discussion section where in line 401 the authors mention the difficulty of benchmarking. One common problem with benchmarking of ML-based generation is the metrics used for benchmarking are underspecified, which limits their utility and can lead to design that maximize the metrics, but are not useful for downstream applications. Some useful references include [1],[4],[5].

[1] Ghugare, Raj, et al. "Searching for High-Value Molecules Using Reinforcement Learning and Transformers." The Twelfth International Conference on Learning Representations.

[4] Steshin, Simon. "Lo-hi: Practical ml drug discovery benchmark." Advances in Neural Information Processing Systems 36 (2023): 64526-64554.

[5] Jain, Moksh, et al. "Multi-objective gflownets." International conference on machine learning. PMLR, 2023.

Response 1.2 – We thank the reviewer for their suggestion. We have modified the discussion section to include these considerations:

"Deep dreaming is just one of several emerging models in the field of inverse MOF design, each varying in their core methodologies, data requirements for training and validation, and the realism of their predictions. This diversity prompts a practical question: which model should be used, and how can their performance be compared? Answering this is complicated by the challenge of defining robust, consistent

performance metrics that capture the nuances of materials design. For instance, where generative models compete to optimise specific performance metrics, designs may excel according to these metrics yet fail to reflect the complexities of practical materials.^{22,48} Or, the metrics themselves may not be sufficiently expressive to differentiate the performance of one generative model over another. These issues have been highlighted by established benchmarks in areas like drug molecule design, demonstrating the need for more robust metrics that align with real-world applications.⁴⁹

*These challenges become even more pronounced in the realm of inverse MOF design, where a consistent benchmarking framework is notably absent. Currently, performance comparisons are only feasible for a selection of metrics. In **Supplementary Note 10**, we provide a review of other available generative models, including their algorithms, training data requirements, and scope. We then compare the models in terms of decoding validity – which assesses whether a MOF CIF file is correctly constructed from its latent or encoded representations – **and the uniqueness of generated linkers**. We reflect on the advantages and disadvantages of the available models, offering practical guidance for their selection. **Here, however**, we would like to emphasise the limited scope of **these metrics** in assessing the utility of a generative model. For example, while decoding validity is useful for comparison, it does not capture important practical considerations, such as the synthesizability of MOFs generated *in silico*.” – Lines 397-417*

*“It is clear from this case study that the SC score can be used to mitigate some of the linker feasibility concerns, yet it still remains a proxy to more rigorous definitions of MOF synthesizability. **To compliment generative modelling outcomes**, [. . .] Feasibility in these cases often rely on expert chemical intuition rather than forming an integral part of the property-driven optimisation, which can limit the practical impact of generative modelling results. **Evidently, more work is needed to arrive at a set of indicators that effectively probe the performance of generative models suitable for downstream MOF applications.**” – Lines 445-457*

Reviewer #1 (Remarks on code availability):

I appreciate the additions to the code repo.

Reviewer #2 (Remarks to the Author):

Remark – Authors have sufficiently answers my comments and the manuscript may be accepted for publication. I do not have further questions.

Response – We thank the reviewer for their original suggestions. We are pleased the manuscript is now to the reviewer's satisfaction.